# A 50 Hz magnetic field affects hemodynamics, ECG and vascular endothelial function in healthy adults: A pilot randomized controlled trial

Hideyuki Okano[1]*, Akikatsu Fujimura[2], Tsukasa Kondo[2], Ilkka Laakso[3], Hiromi Ishiwatari[4], Keiichi Watanuki[1,2,5]

1 Advanced Institute of Innovative Technology, Saitama University, Saitama, Japan, 2 Graduate School of Science and Engineering, Saitama University, Saitama, Japan, 3 Department of Electrical Engineering and Automation, Aalto University, Espoo, Finland, 4 Soken Medical Co., Ltd., Tokyo, Japan, 5 Brain and Body System Science Institute, Saitama University, Saitama, Japan

* hideyukiokano@aol.com

**Data Availability Statement:** All relevant data are within the paper and its Supporting Information files.

## Abstract

Application of exposure to 50/60 Hz magnetic fields (MFs) has been conducted in the treatment of muscle pain and fatigue mainly in Japan. However, whether MFs could increase blood flow leading to muscle fatigue recovery has not been sufficiently tested. We investigated the acute effects of a 50 Hz sinusoidal MF at $B_{max}$ 180 mT on hemodynamics, electrocardiogram, and vascular endothelial function in healthy young men. Three types of regional exposures to a 50 Hz MF, i.e., forearm, upper arm, or neck exposure to MF were performed. Participants who received three types of real MF exposures had significantly increased ulnar arterial blood flow velocity compared to the sham exposures. Furthermore, after muscle loading exercise, MF exposure recovered hemoglobin oxygenation index values faster and higher than sham exposure from the loading condition. Moreover, participants who received real MF exposure in the neck region had significantly increased parasympathetic high-frequency activity relative to the sham exposure. The MF exposure in the upper arm region significantly increased the brachial artery flow-mediated dilation compared to the sham exposure. Computer simulations of induced *in situ* electric fields indicated that the order-of-magnitude estimates of the peak values were 100–500 mV/m, depending on the exposure conditions. This study provides the first evidence that a 50 Hz MF can activate parasympathetic activity and thereby lead to increase vasodilation and blood flow via a nitric oxide-dependent mechanism.

**Trial registration:** UMIN Clinical Trial Registry (CTR) UMIN000038834. The authors confirm that all ongoing and related trials for this drug/intervention are registered.

## Introduction

Many experimental and epidemiological studies for exploring biological and health effects on extremely low-frequency magnetic fields (ELF-MFs) ranging 1–300 Hz have been carried out

**Funding:** This study represents independent research part funded by the Advanced Institute of Innovative Technology, Saitama University. The funders had no role in study design, data collection and analysis, decision to publish, or preparation of the manuscript. There was no additional external funding received for this study.

**Competing interests:** HI is a president and a major shareholder of Soken Co., Ltd., the manufacturer of the magnetic field exposure device used in the present study. The magnetic field exposure device described in this manuscript was provided by Soken Co., Ltd. The scientific content of this work was not directly supported by Soken Co., Ltd. All other authors declare not to have any conflict of interest. This does not alter the authors' adherence to PLOS ONE policies on sharing data and materials.

**Abbreviations:** MF, magnetic field; EF, electric field; FEM, finite-element method; fNIRS, functional near infrared spectroscopy; oxyHb, oxyhemoglobin; deoxyHb, deoxyhemoglobin, totalHb, total hemoglobin; ECG, electrocardiogram; RRI, R-R interval; HF, high-frequency; LF, low-frequency; HRV, heart rate variability; FMD, flow-mediated dilation; NO, nitric oxide.

and reviewed [1–51]. Aside from potential health concerns, medical applications of ELF-MFs are promising research areas to be explored using a ubiquitous sinusoidal frequency. However, the literature on the effects of sinusoidal MFs on cells and tissues is not as abundant as studies on pulsed MFs [1] and the underlying action mechanisms of sinusoidal MFs for medical therapeutic applications remain poorly understood [2–4]. Interestingly, several preclinical studies in cells and animals have suggested that 50/60 Hz sinusoidal MFs ranging 0.4–5 mT (rms) can improve wound repair [5, 6], osteogenesis [7, 8], and neurogenesis [8–12]. In addition, a 60 Hz MF at 6 mT promoted cell proliferation in Hela and IMR-90 cells by decreasing intracellular reactive oxygen species (ROS) levels [13]. A 50 Hz MF of 1 mT induced a proliferative and survival advantage even in malignant tumor cells by activating antioxidative and detoxification cytoprotective pathways [14]. In contrast, 50 Hz MFs ranging 0.9–4.8 mT inhibited osteoblast proliferation in a non-dose-dependent manner [15], and a 60 Hz MF at 1.5 mT inhibited epidermal keratinocyte proliferation [16]. Other studies reported that 50/60 Hz MFs ranging 0.5–3 mT did not affect either cell proliferation or cell cycle [17, 18]. These inconsistent and contradictory experimental results may be partly explained by the large variation in experimental variables between studies with respect to waveform, frequency, amplitude, duration, cell type, cell age and treatment [19–21]. If the consistent and reasonable results of a specific exposure condition are obtained using 50/60 Hz sinusoidal MFs, they help us to better understand the interaction mechanisms between 50/60 Hz sinusoidal MFs and biological systems.

When mainly focusing on Japan, non-thermal noninvasive 50/60 Hz sinusoidal MF therapy has been used to relieve chronic pain, muscle stiffness, muscle fatigue and so on, for over 40 years, since approved by Japanese Ministry of Health, Labour and Welfare for improvement of muscle stiffness and blood circulation in the MF-exposed region, in which the peak magnetic flux density $B_{max}$ at the surface area of the approved therapeutic MF exposure devices should range between 35 and 180 mT. In some studies, head exposure to 50/60 Hz MFs in 3–50 mT range altered electroencephalogram (EEG) signals in several brain regions, indicating the presence of magnetosensory evoked potentials [22, 23]. In heart exposure, 50 Hz MFs of 150–200 µT for 30 min significantly reduced low-frequency parameters of heart rate variability (HRV) through known mechanisms [24]. In short-term exposure to 50 Hz MF intensity of up to 100 µT, however, no significant change was detected in HR and HRV [25]. In addition, a recent clinical study has shown that application of MF therapy under exposure to a 50 Hz MF of 10 mT for 20 min significantly reduces pain symptoms and leads to an improvement of functional ability in patients with low back pain, based on the analysis of both subjective parameters, such as visual analogue scales (VAS), and objective parameters reflecting the range of spinal mobility [26]. However, the clinical evidence examined using objective parameters, such as physiological monitoring measures, is not sufficient to support the effectiveness of this therapeutic approach, especially due to the MF-enhanced blood circulation and recovery of muscle fatigue and pain [27–31].

In this study, we attempted to clarify the acute physiological effects of a 50 Hz sinusoidal MF exposure at peak magnetic flux density $B_{max}$ 180 mT or $B_{rms}$ 127 mT on arterial blood flow velocity, blood pressure, heart rate, blood oxygenation (concentration of oxyhemoglobin [oxyHb], deoxyhemoglobin [deoxyHb] and total hemoglobin [totalHb]), electrocardiogram (ECG) and flow-mediated dilation (FMD), an index of artery endothelium-derived nitric oxide (NO)-mediated vasodilator function, in healthy human subjects.

## Methods

This study adhered to the Declaration of Helsinki, and was approved by the Ethics Committee of the Saitama University. All study procedures were approved by the Institutional Review Board

(IRB) of the Saitama University. IRB approval number: H29-E-13. Date of IRB approval: 24/10/ 2017. This study was registered with the UMIN Clinical Trials Registry (CTR): UMIN000038834. Date of protocol fixation: 07/02/2017. Anticipated trial start date: 25/10/2017. Full date of registration: 09/12/2019. The full trial protocol is available at https://upload.umin.ac.jp/cgi-open-bin/ctr_ e/ctr_view.cgi?recptno=R000044252. The trial protocol is also given as S1 File (S2 File in Japanese). To conduct the trial, ethical IRB approval was obtained; however, registration to UMIN-CTR could not be achieved by the start of recruitment and randomization of participants. We noticed after the study had been partially completed that UMIN-CTR accepted trial registration even after participant randomization. Hence, we have agreed with the significance of UMIN-CTR and registered this study. Thus, trial registration occurred during the recruitment phase. The late registration was due to an error of omission. Such late registration does not affect study results and participants. We confirmed that all ongoing and related trials for this intervention are registered. We hereby state that all future trials will be registered prospectively.

The clinical trial was conducted at Advanced Institute of Innovative Technology, Saitama University between October 25, 2017 and April 3, 2020. This study was conducted in accordance with the Consolidated Standards of Reporting Trials (CONSORT) 2010 statement. The CONSORT checklist is given as S3 File. All participants signed informed consent before any study procedures were performed. The informed consent form is given as S1 File (S2 File in Japanese).

## Participants

First, 18 healthy right-handed adult male volunteer students were recruited by e-mail from Saitama University. We asked them to participate in face-to-face interview. A response was obtained from 14 men, of whom 12 were eligible and then enrolled in protocol A including the measurements of blood flow velocity, blood pressure and heart rate, the functional near infrared spectroscopy (fNIRS) measurement, and the ECG measurement (Fig 1). The inclusion criteria were as follows: 1) participants have not used any form of physical therapy and have not taken any vasoactive medication during the study period; 2) subjects' body temperature, heart rate (HR), systolic and diastolic blood pressures (BP) were within normal ranges. Two subjects were excluded as they did not meet inclusion criteria (HR > 100 bpm, systolic BP > 130 mmHg or diastolic BP > 85 mmHg) (Fig 1). Eventually, recruitment and enrollment of 10 participants in protocol A was conducted from October 2017 to April 2020. Thus, protocol A was completed in 10 participants (Fig 1).

Second, 16 healthy right-handed adult male volunteer students were recruited and enrolled in protocol B for FMD measurement alone using the same criteria (Fig 1). Protocol B included 6 participants from protocol A. Recruitment and enrollment in protocol B was conducted from February 2018 to March 2020.

In both protocol A and B conditions, subjects' body temperatures were confirmed to be within normal ranges using an electronic clinical thermometer (MC-681, OMRON Healthcare Co., Ltd., Kyoto, Japan), and systolic and diastolic BP values and HR values were confirmed to be within normal limits using a digital sphygmomanometer with an upper arm cuff (DSK-1051J, NISSEI Co., Ltd., Tokyo, Japan). There were no significant differences between the subject baseline characteristics in the protocols A and B (mean ± SD, $n = 10$ in A condition and $n = 16$ in B condition) as shown in Table 1.

## Study design

The present study was designed as a prospective, randomized, double-blind, sham (placebo)-controlled, counterbalanced, crossover trial. The values of blood flow velocity, BP, HR, blood oxygenation (concentration of oxyHb, deoxyHb and totalHb), ECG parameters, and FMD

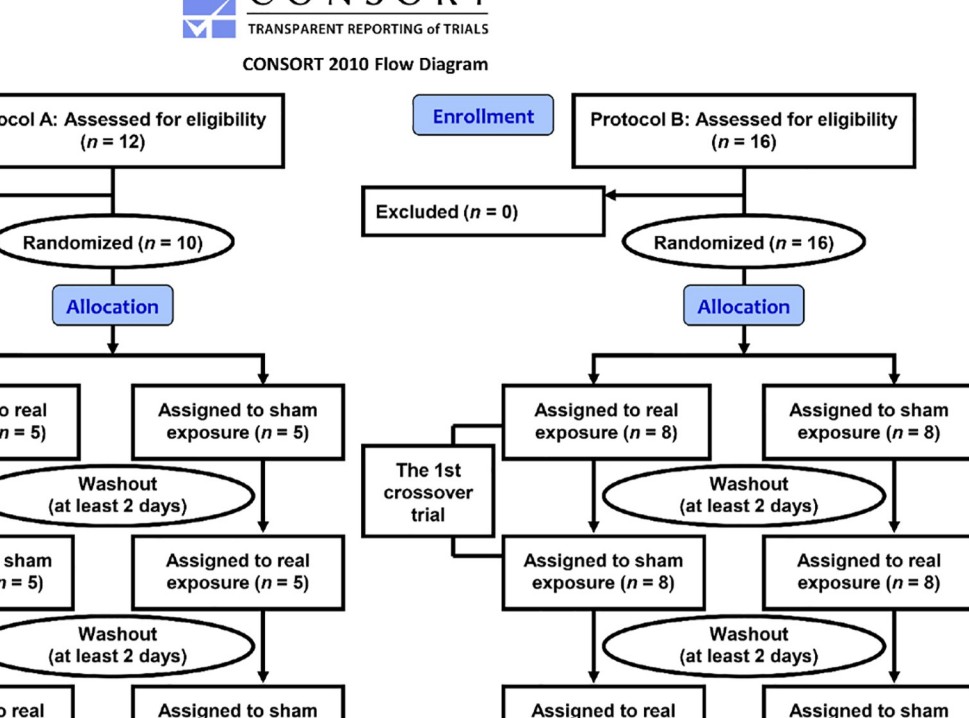

**Fig 1. Flow chart of the study.** The flow chart illustrates the basic layout of the clinical trial including steps involved in recruitment of study participants and the actual steps in the clinical trial of protocols A and B.

were compared with two different exposure conditions, i.e., real MF and sham control exposures. Participants who received either a real MF or a sham exposure and technicians who monitored the outcomes were blinded to real MF or sham exposure, which was randomized between crossover trials. Each crossover trial for each participant was performed twice on different days as the first and second trials after a washout period of at least two days across the four testing sessions (Fig 1), and the duplicate data of real MF or sham exposure were averaged for each individual in each test. The initiation of measurements in all trials was done after more than 10-min rest with a sitting position. All trials were performed during daytime (11:00–17:00) at room temperature (25˚C ± 0.5˚C) and relative humidity of 50% ± 10%.

## Blood flow velocity

We made the preliminary trial protocol of measurement of ulnar arterial blood flow velocity under forearm exposure to a 50 Hz MF and the protocol was described previously [50]. In this

**Table 1. Subject baseline characteristics in the protocols A and B.**

| Measured outcomes | Protocol A (*n* = 10) | | Protocol B (*n* = 16) | | *p* |
|---|---|---|---|---|---|
| | Mean ± SD | Range | Mean ± SD | Range | |
| Age (y.o.) | 22.8 ± 0.4 | 22–23 | 23.2 ± 1.9 | 21–27 | 0.7593[b] |
| Height (cm) | 171.1 ± 5.1 | 164.3–179.8 | 171.0 ± 3.7 | 165.4–180.3 | 0.9622[a] |
| Weight (kg) | 59.5 ± 8.0 | 49.7–72.1 | 59.2 ± 7.6 | 49.4–68.6 | 0.9387[a] |
| Body mass index (kg/m$^2$) | 20.3 ± 2.4 | 17.3–24.1 | 20.3 ± 2.6 | 17.1–24.1 | 0.7919[b] |
| Systolic BP (mmHg) | 117.6 ± 4.9 | 112–125 | 120.3 ± 5.1 | 112–129 | 0.1999[a] |
| Diastolic BP (mmHg) | 70.7 ± 5.7 | 63–80 | 71.5 ± 6.3 | 58–83 | 0.7402[a] |
| HR (bpm) | 66.2 ± 6.5 | 58–80 | 67.3 ± 7.3 | 52–80 | 0.6897[a] |
| Body temperature (°C) | 36.7 ± 0.2 | 36.4–36.9 | 36.7 ± 0.2 | 36.4–36.9 | 1.0000[b] |

Mean ± SD, *n* = 10 in A condition and *n* = 16 in B condition. Protocol B included 6 participants from protocol A. The Shapiro-Wilk test was performed to assess data distribution (Table A in S4 File). In the case of parametric distribution, *p*-values between experimental conditions were calculated with [a]the Student's *t*-test. In the case of nonparametric distribution, these were obtained with [b]the Wilcoxon rank-sum test.

study using the protocol as the primary outcome measure, we measured blood flow velocity under three different regional exposures to a 50 Hz MF. The protocol is shown in Fig A in S4 File. The experiments were performed in the sitting position, remaining in constant contact with an MF exposure device. The measurement of blood flow velocity was conducted under one of three types of regional exposures, i.e., forearm, upper arm, or neck exposure to a 50 Hz MF at $B_{max}$ 180 mT or $B_{rms}$ 127 mT on the surface of the MF exposure device. The MF-exposed side of the forearm and upper arm was the dorsal side of the arm and that of the neck was the back side of the neck. The MF or sham exposure was performed continuously for 15 min. In all cases of exposures, an ultrasound probe was positioned on the ventral skin surface of the bared left forearm to assess the blood flow velocity in an ulnar artery. In the case of the forearm exposure, after putting on the dorsal side of the left forearm on the MF exposure device for about 5 min, the blood flow velocity (peak systolic velocity [PSV] in pulsed-wave [PW] mode) in an ulnar artery was measured noninvasively from the ventral side of the forearm at 5-min intervals for 25 min using a portable digital color Doppler ultrasound system (JS2 with a linear ultrasound probe L741, 5–12 MHz, Medicare Co., Ltd., Shenzhen, China). The MF exposure device was covered with a specially designed thin cover cloth (P27Mar15, Soken Co., Ltd., Toride, Japan), which is made of cotton to absorb perspiration.

The procedures for measuring the blood flow velocity have been reported in detail elsewhere [50]. Briefly, when holding the left arm at 120–140° of elbow flexion with holding the palm upward, the forearm was positioned on an MF exposure device for about 30 min in order to keep the arm immobile still as long as possible during the clinical trial in each subject. At each time point, the PSV was measured in triplicate for the blood flow velocity and the mean values were calculated. For the upper arm exposure, an additional second MF exposure device was positioned to the dorsal side of the left upper arm. For the neck exposure, an additional third MF exposure device was positioned laterally to the dorsal side of the neck. In the cases of the upper arm and neck exposure experiments, the MF exposure device for the forearm exposure was used to keep the same posture as in the case of the forearm exposure experiment without turning on the power. Each crossover trial for each participant was performed twice on different days after a washout period of at least two days and the duplicate data were averaged for each individual in each test (Fig 1). Therefore, the mean values of two trials in MF or sham exposure were used for statistical analyses.

## Blood pressure and heart rate

Arterial blood pressure (BP) and heart rate (HR) monitoring were performed under the forearm exposure alone. The MF or sham exposure was conducted continuously for 15 min. Systolic and diastolic BP and HR values were measured at 10-min intervals of the pre-exposure (baseline), during the exposure (at 10 min after the start of the exposure), and the post-exposure (at 5 min after the termination of the exposure) using a digital BP monitor with an upper arm cuff (DSK-1051J, NISSEI Co., Ltd., Tokyo, Japan) under the same experimental condition of the forearm exposure as one of the secondary outcome measures twice on different days from the measurement of blood flow velocity after a washout period of at least two days and the duplicate data were averaged for each individual in each test (Fig 1).

## fNIRS

fNIRS is an optical imaging modality that exploits near infrared light ranged between (700–900 nm), and the relative concentration of hemoglobin was calculated by applying the modified Lambert-Beer law [52]. Regional skeletal muscle blood flow can be noninvasively measured using NIRS [53, 54]. The changes of muscle blood oxygenation were monitored in the ventral side of the left forearm at evaluation points, i.e., middle part of flexor carpi radialis muscle at channel (CH)-1 and proximal part of the muscle at CH-2 due to the higher response to muscle loading exercise. The monitoring of muscle blood oxygenation was conducted in a sitting position. During the entire experiments, the concentration of ΔoxyHb, ΔdeoxyHb, and ΔtotalHb in two muscle regions (CH-1 and CH-2) was measured consecutively at sampling frequency of 1 Hz (1 s per data point) using a compact and wearable multichannel fNIRS device (Pocket NIRS Duo, DynaSense, Hamamatsu, Japan). In this study, the hemoglobin oxygenation index (HOI) values (ΔoxyHb–ΔdeoxyHb concentration) [55, 56] were calculated and compared between the MF- and sham-exposed groups.

Wrist curl training method [57] for muscle loading exercise was selected to examine the condition of the wrist flexor muscles in the left forearm (in the case of a right-handed person). The protocol of muscle loading exercise is shown in Fig B in S4 File. Briefly, the participants performed the wrist curl exercise using a 3kg dumbbell (SINTEX Chrome array 3kg STW021, Sinwa Enterprise, Osaka, Japan) for 5 min in a chair sitting position. Each participant held the dumbbell, kept the forearm stationary, turned the palm up, and raised the wrist freely, then raised the dumbbell to the dorsiflexion position of the wrist joint in 1 s. Thereafter, the dumbbell was lowered to the maximum range of the wrist flexion from the dorsiflexion position in 2 s. It took 3 s to perform 1 cycle. The participant conducted 20 repetition cycles in 1 min and after that took rest for 1 min. Thus, a 5-min exercise consisted of 3 sets of 20 repetition cycles with a 1-min interval between sets.

For the MF exposure, the left forearm was exposed to 50 Hz MF for 15 min immediately after cessation of the muscle loading exercise (Fig B in S4 File). The dorsal side of the left forearm was positioned on a 50 Hz MF exposure device for 25 min (15-min exposure and 10-min post-exposure periods) to keep the hand and arm motionless as long as possible after the muscle loading exercise. Each crossover trial for each participant was performed twice on different days after a washout period of at least two days and the duplicate data were averaged for each individual in each test (Fig 1).

## ECG

The ECG monitoring is shown as one of the secondary outcome measures under the neck exposure to a 50 Hz MF in Fig C in S4 File. The ECG monitoring was conducted under the neck exposure alone. The MF exposure device for the forearm exposure was also used to keep

posture the same as in the case of the forearm exposure experiment without turning on the power. Therefore, the subjects' postures in the ECG monitoring were the same as those for the measurement of blood flow velocity. The ECG monitoring in each subject was carried out on different days with the measurement of blood flow velocity. A wireless ECG electrode was attached on the skin surface of the chest over the heart using medical tape. After attaching the ECG electrode for about 5 min, ECG was recorded continuously for 25 min using a multi-channel telemetry system (WEB-1000, Nihon Kohden Co., Ltd., Tokyo, Japan). The MF or sham exposure was conducted continuously for 15 min. Each crossover trial for each partici-pant was performed twice on different days after a washout period of at least two days and the duplicate data were averaged for each individual in each test (Fig 1).

The R-R interval (RRI) of ECG signal were extracted and used in this analysis. A power spectral analysis of RRI by fast Fourier transformation (FFT) was performed to obtain the low-frequency (LF) and high-frequency (HF) power components of HRV over an experimental period of 25 min. The procedures for determining the power-spectral analysis of RRI have been reported in detail elsewhere [58]. Briefly, the frequency range of 0.05–0.15 Hz was com-puted as the LF component of HRV, which is an index of both sympathetic and parasympa-thetic activities [59]. The frequency range of 0.15–0.40 Hz was computed as the HF component of HRV, which reflects mainly parasympathetic activity (vagal tone) [60]. In con-trast, the LF/HF ratio reflects mainly sympathetic activity [61].

## FMD

FMD provides an index of artery endothelium-dependent NO-mediated vasodilator function and the FMD measurement has been reported elsewhere [62–64]. Briefly, an FMD monitoring device equipped with a 10 MHz H-type ultrasound probe (UNEXEF 18VG, UNEX Co., Ltd., Nagoya, Japan) has a high-resolution linear artery transducer coupled to computer assisted analysis software used in an automated edge detection system for the measurement of the bra-chial artery diameter (% change from the baseline diameter).

We conducted the preliminary trial protocol of FMD measurement of the left brachial artery under the left upper arm exposure to a 50 Hz MF and the protocol was described previ-ously [51]. FMD measurement was performed as shown in Fig D in S4 File. An occlusion cuff was placed around the forearm in accordance with the user manual (Fig D in S4 File). The bra-chial artery was automatically scanned longitudinally 5–10 cm above the elbow (Fig D in S4 File). When the clearest B-mode image of the intima–media complex was obtained, the trans-ducer was held at the same point throughout the scan by a stereotactic probe holder to ensure consistency of the imaging. Depth and gain settings were adjusted as necessary to optimize the images of the arterial lumen wall interface. When the tracking gate was placed on the intima, the artery diameter was automatically tracked, and the waveform of diameter changes over the cardiac cycle was displayed in real time using the FMD mode of the tracking system. This allowed the ultrasound images to be optimized at the start of the scan and the transducer posi-tion to be adjusted immediately for optimal tracking performance throughout the scan. Pulsed Doppler flow was assessed at the baseline and during peak hyperemic flow, which was con-firmed to occur within 15 s after cuff deflation (decompression). Blood flow velocity was calcu-lated from the Doppler data and displayed as a waveform in real time. The baseline longitudinal images of the artery were acquired for 30 s, and then the occlusion cuff was inflated to 50 mmHg above systolic pressure for 5 min. The longitudinal images of the artery were automati-cally recorded continuously until 2 min after cuff deflation. Pulsed Doppler velocity signals were obtained for 20 s at the baseline and for 10 s immediately after cuff deflation. Changes in brachial artery diameter were immediately expressed as a percentage change relative to the

vessel diameter before cuff inflation. The FMD value was obtained by the following equation:

$$\text{FMD (\%)} = \frac{D1 - D2}{D1} \times 100 \tag{1}$$

where $D1$ = the basal diameter and $D2$ = the maximum diameter reached after cuff release. Thus, FMD values were automatically calculated as the percentage change in peak vessel diameter from the baseline value.

Exposure of the dorsal side of the left upper arm to a 50 Hz MF at $B_{max}$ 180 mT or $B_{rms}$ 127 mT was performed with the subject lying on a bed in the supine position for the FMD measurement as one of the secondary outcome measures. The brachial artery FMD values in the left upper arm were monitored during the pre-exposure (baseline) and the post-exposure. After putting on the dorsal side of the left upper arm on an MF exposure device and attaching the cuff around the left forearm about at least 5-min resting period, it took about 10 min to obtain FMD values per one measurement (Fig D in S4 File). The FMD values were monitored before the exposure and immediately after the termination of the exposure (Fig D in S4 File). From a physiological point of view, in the case of repeated measurements of FMD, a time interval of measurement is recommended at least 30 min and accordingly FMD test was frequently performed before and after 30-min interventions to examine the acute effects of interventions and to minimize repeated assessment effects [62–64]. Therefore, FMD test was repeated with an interval of 30 min and the MF or sham exposure was conducted continuously for 30 min between tests. Each crossover trial for each participant was performed twice on different days after a washout period of at least two days and the duplicate data were averaged for each individual in each test (Fig 1).

## MF exposure

In the active 50 Hz MF condition, participants were exposed to a 50 Hz MF for 15 min (protocol A) or 30 min (protocol B) using an AC MF exposure device (Soken MS, Soken Co., Ltd., Toride, Japan). The MF exposure device was utilized for research purpose. Two separate electromagnetic coils (inner diameter 20 mm, outer diameter 50 mm, and height 60 mm) are set horizontally inside the MF exposure device and the distance between the centers of the double induction coils is 160 mm. The two coils were electrically connected in series.

The measurement of root mean square (rms) values ($a$) was made by means of a single-axis teslameter with a suitable transverse probe (F41 teslameter, FP-2X-250-TS15 probe, Lake Shore Cryotronics Co., Westerville, OH, USA). The $B_{max}$ values ($b$) of a 50 Hz MF were calculated from the measured $B_{rms}$ values ($a$) as:

$$b = \sqrt{2}a \tag{2}$$

The value of $B_{max}$ is 180 mT on the surface of the MF exposure device above the centers of the coils (Fig E in S4 File). The $B_{max}$ values decrease exponentially with distance. The estimated $B_{max}$ values in an ulnar artery, a brachial artery, and a carotid artery are approximately 13 mT, 8 mT, and 5 mT, in which the distances from the surface of the MF exposure device at $B_{max}$ 180 mT are approximately 3 cm, 4 cm and 6 cm, respectively (Fig F in S4 File). The estimated $B_{max}$ value in the heart was not detected using this magnetometer because the distance is more than 10 cm between the MF exposure device and the heart.

The surface temperature of the MF exposure device during the MF exposure period of up to 30 min was maintained at 25°C ± 0.5°C using a laptop fan cooler (OPOLAR LC06, OPOLAR Co., Sugar Creek, MO, USA). The relative humidity was controlled at 50% ± 10%. No vibration was detected during the MF exposure period using a horizontal checking device (PWM-25-100, KOD Co., Ltd., Ayabe, Japan). Ambient room noise was 50–55 dB and the

sound pressure level of the device was below these levels, which were measured using a sound meter (LA-1440, Ono Sokki Co., Ltd., Yokohama, Japan).

## Sample size

The required sample size per group was based on the preliminary estimates from the results of blood flow velocity of 10 subjects under forearm exposure to a 50 Hz MF for 15 min (Fig 2). The blood flow velocity values of all subjects in the sham-exposed group decreased during 15-min sham exposure period, and mean (SD) decrease of ten subjects was 7.7 cm/s (64.8 [22.1] cm/s to 57.1 [22.2] cm/s) (Fig 2A). In contrast, the blood flow velocity values of all subjects in the MF-exposed group increased during 15-min MF exposure period, and mean (SD) increase of 10 subjects was 6.9 cm/s (66.0 [17.6] cm/s to 72.9 [18.3] cm/s) (Fig 2B). From these results, mean (SD) values of blood flow velocity differences between the MF- and sham-exposed groups (MF–sham) gradually increased: baseline, 1.3 (9.9) cm/s; 5 min, 8.6 (9.0) cm/s; 10 min, 15.1 (10.6) cm/s; 15 min, 15.5 (10.0) cm/s (Fig 2C).

Here, to detect a statistical difference in the mean change from the baseline between real MF and sham control exposures, we estimated the magnitude of the MF effects on blood flow velocity by setting SD = 10 cm/s and power = 0.80 using a statistical software package (EZR version 1.41, Saitama Medical Center, Jichi Medical University, Saitama, Japan), which is available online (http://www.jichi.ac.jp/saitama-sct/SaitamaHP.files/statmedEN.html). The results of sample size requirements are shown in Table 2.

If the MF exposure is expected to increase blood flow velocity by 13 cm/s relative to the sham exposure, 10 subjects would be expected to find statistical significance as the minimum sample size per group. Our obtained results of mean (SD) values of blood flow velocity differences between both groups are approximately 15 (10) cm/s for 10–15-min exposure, and in this case, the estimated sample size required to determine significant MF effects is 7 subjects per each group. For the primary outcome measure, the estimated minimum sample size of 10 subjects would be able to detect significant differences between both groups. Therefore, on the basis of the significant results from blood flow velocity of 10 subjects per condition, we decided from the beginning that the target number of subjects was set at 10 per condition as a pilot test for all outcome measures for protocol A. Subsequently, after analyzing the FMD results obtained from 10 subjects, we estimated the sample size required to determine significant MF effects, and it is estimated that at least 14 subjects would be required to reach statistical power of 80%. Therefore, an additional 6 subjects were enrolled in protocol B.

## Randomization

Randomization was performed according to our preliminary trial protocol [50, 51]. Briefly, all participants were divided evenly into two conditions, a real MF (A) and a sham (B) exposures, with an allocation ratio of 1:1 after block randomization with a permuted block size of 2 according to computer-generated random numbers by an administrative controller/investigator (H.O.) Thus, the allocation of which condition would receive either a real MF (A) or a sham (B) exposure for the first time was randomly generated. When considering the case of the flow chart shown in Fig 1, 2-treatment and 4-period crossover design (2 × 4 crossover sequence) was done; ABAB or BABA.

## Blindness

The allocation was blinded to both the participants and the technicians. Real MF and sham control exposures were also blinded to both of them except for one investigator who was regarded as an administrative controller/investigator (H.O.) of an MF exposure device who

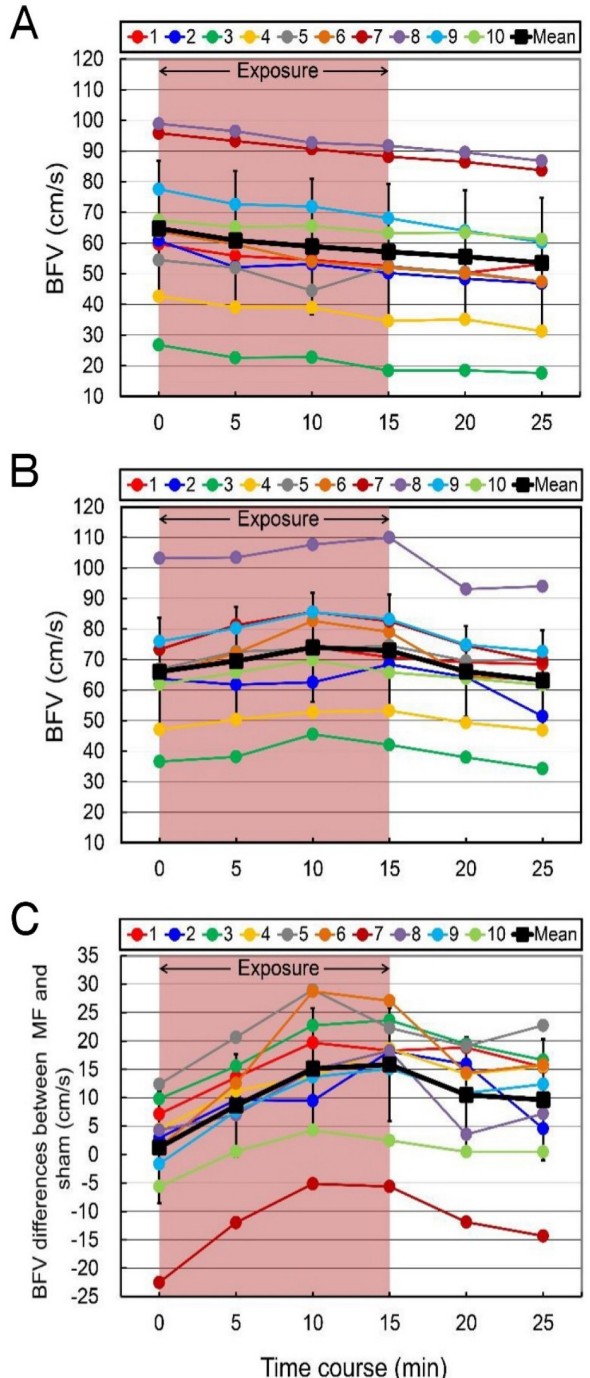

**Fig 2. The time course of ulnar artery blood flow velocity (BFV) in forearm exposure to a 50 Hz MF.** (A) Sham exposure. (B) MF exposure. (C) MF exposure - sham exposure. The duration of exposure is 15 min. The numbers indicate the individual identification numbers.

did not have any contact with the participants and technicians. The operation switch was turned on and off remotely by the investigator using an extension cord. Technicians were not allowed to check the switch of the MF exposure device. All participants were not informed as to when the MF exposure device was switched on or off.

**Table 2. Sample size required to determine significant 50 Hz MF effects on blood flow velocity (BFV).**

| BFV difference (cm/s) | No. of subjects |
| --- | --- |
| 5.0 | 63 |
| 6.0 | 44 |
| 7.0 | 33 |
| 8.0 | 25 |
| 9.0 | 20 |
| 10.0 | 16 |
| 11.0 | 13 |
| 12.0 | 11 |
| 13.0 | 10 |
| 14.0 | 9 |
| 15.0 | 7 |

## Statistical analyses

Statistical analyses were performed using SAS Analytics version 9.4 (SAS Institute, Cary, NC, USA). A two-way repeated-measures ANOVA was calculated with the factors "Time" and "Condition". *Post hoc* analysis of differences between the MF and sham exposures was made with the Student's *t*-test or the Wilcoxon rank-sum test (between conditions), and analysis of within the same exposure was performed with the paired *t*-test or the Wilcoxon signed-rank test (within a condition) with two-sided tests, according to the normality of samples' distribution verified with the Shapiro-Wilk test. For all comparisons, $p < 0.05$ was considered significant.

## Dosimetry

The *in situ* electric fields (EFs) induced by a 50 Hz MF generated by double coils were estimated computationally for the left forearm, left upper arm, and neck exposures. Computational methods for determining the MF-induced EFs in heterogeneous anatomically realistic body voxel models are based on the quasi-static approximation of Maxwell's equations. Under the quasi-static approximation, the induced EFs can be represented in terms of the electric scalar potential which satisfies an elliptic partial-differential equation. The EFs were numerically calculated using the finite-element method (FEM) [65, 66] with first-order cubical elements [67, 68]. Based on the measurement of a 50 Hz MF, the MF exposure coils were modeled using a collection of thin circular current loops. For modeling the anatomy of the torso and arm, a voxel model with a resolution of 0.5 mm was constructed based on the BodyParts3D database (http://lifesciencedb.jp/bp3d/) and is available online (http://version.aalto.fi/ilaakso/alvar). The voxel model includes detailed models of vasculature and consists of 42 different tissues and organs. The electrical conductivity values of tissues at 50 Hz were identical to those listed in the previous study [33]. To reduce numerical artifacts, the voxels with the highest 0.01%-induced EF strength in each tissue were excluded from the analysis.

## Results

Twenty-six participants in total were assessed in duplicate measurements on different days for crossover MF and sham exposure experiments (Fig 1). Protocol B included 6 participants from protocol A. Therefore, 20 individuals randomly participated in either protocol A or B, and 6 individuals of them participated in both protocols. All participants had neither discomfort nor

any sensation including heat and magnetic sensation during the study period and they have not dropped out after participating in each trial.

## Normality tests

The Shapiro-Wilk test was run to test the assumptions of normality. As expected, the results revealed that most of the data was normally distributed (Table A in S4 File) and in the case of parametric distribution, $p$-values were calculated with the paired $t$-test (within a condition) and the Student's $t$-test (between conditions). However, some of the data was not normally distributed (Table A in S4 File) and in the case of nonparametric distribution, these were obtained with the Wilcoxon signed-rank test (within a condition) and the Wilcoxon rank-sum test (between conditions).

## Effects of MF exposure on blood flow velocity

The blood flow velocity values were evaluated before (baseline) and after 15-min exposure to MFs at 5-min intervals as described previously [50]. The changing rate (%) from the baseline values of blood flow velocity was analyzed because the variability of the baseline values was very large between individuals, which can be associated with slight body movements over time, depending on the individuals (between-subjects variability). The CV (coefficient of variation = [100% × SD] / mean) values for the baseline values of blood flow velocity in the same individual (within-subject variability), which are an indicator of the reproducibility and accuracy of the method for measuring blood flow velocity, were within 20% across the four testing sessions: 7.0% ± 6.0% (mean ± SD), range 1.1–17.9%, $n = 10$.

The results of the changing rate of blood flow velocity of the ulnar artery in three different regional exposures are shown in Fig 3. The blood flow velocity values in sham exposure were significantly decreased compared with the baseline value (indicated as 100%), probably due to the loading pressure and immobility-induced ischemic conditions by preventing the arm from moving (Table 3A and Fig 3). It is for this reason, for example, that this reduction in blood flow velocity can be induced mainly by the interface pressures (external loading pressures of body weight) on the tissues including blood vessels [69, 70] in combination with being motionless and inactivity [71], and additionally by reduced physiological arousal [29].

In the forearm exposure, the repeated-measures ANOVA showed a significant Condition by Time interaction [$F(5, 90) = 21.51$, $p < 0.0001$]. The MF exposure significantly increased blood flow velocity values during the MF exposure period, and the blood flow velocity value peaked at 10 min after the start of the exposure, increasing from the baseline value by 13.4% (Table 3B and Fig 3A). After the termination of the exposure, the blood flow velocity gradually returned to the baseline value (Fig 3A). These compensatory autonomic responses are protective during the recovery phase of normal cardiovascular homeostasis [60]. The blood flow velocity values were significantly increased by the MF exposure compared with the sham exposure during the exposure and after the termination of the exposure periods (Fig 3A).

In addition, in the upper arm exposure, the repeated-measures ANOVA showed a significant Condition by Time interaction [$F(5, 90) = 11.60$, $p < 0.0001$]. The MF exposure significantly increased blood flow velocity values during the MF exposure period, and the value of blood flow velocity peaked at 5-min, increasing from the baseline value by 4.6% (Table 3C and Fig 3B). The blood flow velocity values were also significantly increased by the MF exposure compared with the sham exposure during the exposure and after the termination of the exposure periods (Fig 3B). The forearm exposure induced more significant increase in the blood flow velocity compared with the upper arm exposure, probably due to the higher MF strength (Fig 3A and 3B).

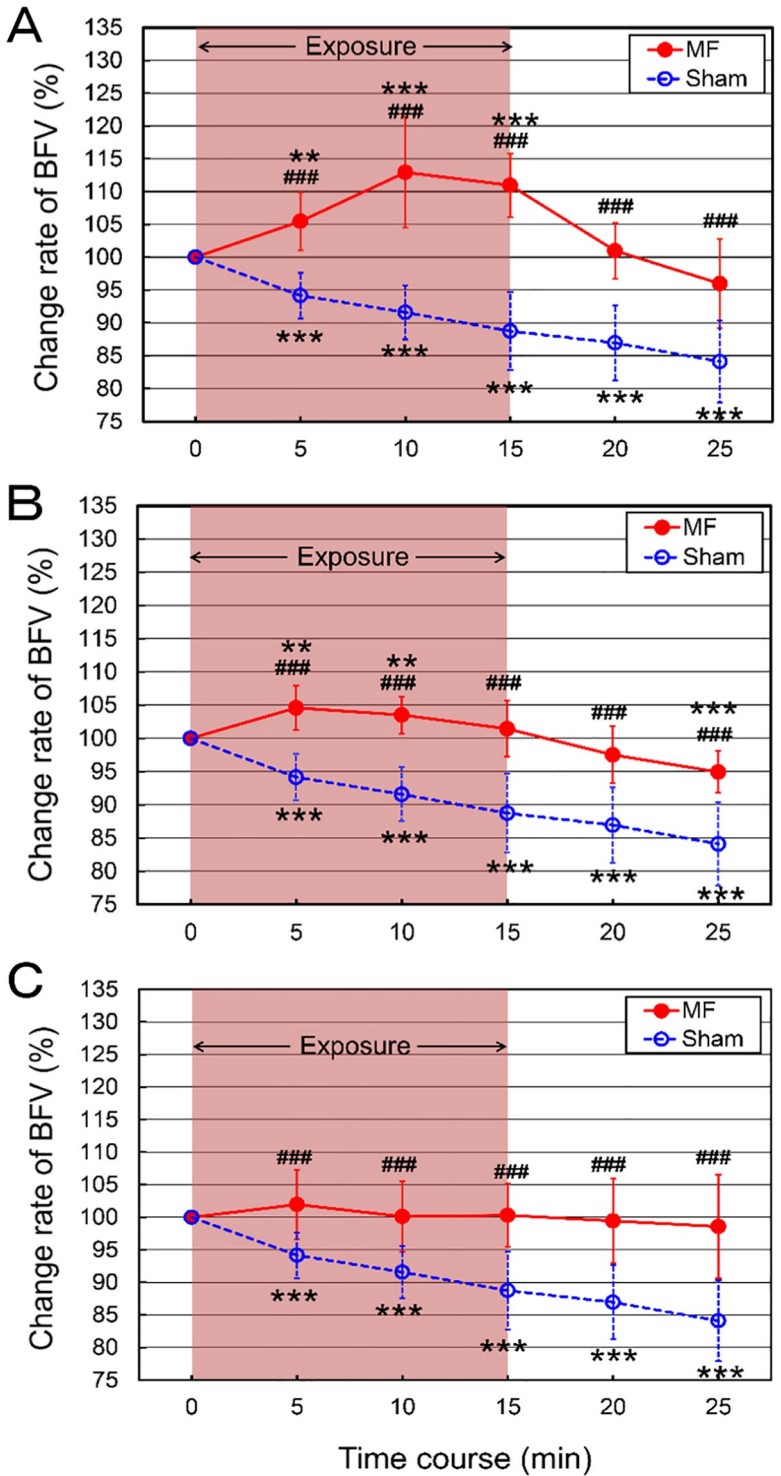

**Fig 3. The time course of change rate (%) in ulnar artery blood flow velocity (BFV) in three different regional exposures to a 50 Hz MF.** (A) Forearm exposure. (B) Upper arm exposure. (C) Neck exposure. The values are expressed as mean ± SD. $n$ = 10 in each condition. The duration of exposure is 15 min. MF exposure, red closed circles; sham exposure, blue open circles. Two-way repeated-measures ANOVA, followed by the Student's *t*-test or the Wilcoxon rank-sum test (between conditions), and the paired *t*-test or the Wilcoxon signed-rank test (within a condition). **$p < 0.01$ compared with the baseline (within a condition). #$p < 0.05$, ##$p < 0.01$, ###$p < 0.001$ compared with the time-matched sham exposure (between conditions).

**Table 3. Comparison of change rate (%) of ulnar arterial blood flow velocity between 50 Hz MF and sham exposures.**

| A. Sham exposure | | | | | | |
|---|---|---|---|---|---|---|
| t (min) | 0 | 5 | 10 | 15 | 20 | 25 |
| **Sham** | 100.00 | 94.15 ± 3.47*** | 91.57 ± 4.06*** | 88.74 ± 5.95*** | 86.94 ± 5.68*** | 84.10 ± 6.25*** |
| | | 0.0005 | 0.0001 | 0.0002 | 0.0000 | 0.0000 |
| **B. Forearm exposure** | | | | | | |
| t (min) | 0 | 5 | 10 | 15 | 20 | 25 |
| **MF** | 100.00 | 105.47 ± 4.40**, ### | 112.94 ± 8.48***, ### | 110.95 ± 4.88***, ### | 100.97 ± 4.30### | 95.96 ± 6.78### |
| | | 0.0034, 0.0000 | 0.0009, 0.0000 | 0.0001, 0.0000 | 0.1934[a], 0.0006[b] | 0.0923, 0.0007 |
| **C. Upper arm exposure** | | | | | | |
| t (min) | 0 | 5 | 10 | 15 | 20 | 25 |
| **MF** | 100.00 | 104.59 ± 3.33**, ### | 103.51 ± 2.79**, ### | 101.45 ± 4.24### | 97.51 ± 4.28### | 94.94 ± 3.16***, ### |
| | | 0.0019, 0.0000 | 0.0032, 0.0000 | 0.3074, 0.0000 | 0.0990, 0.0002 | 0.0007, 0.0003 |
| **D. Neck exposure** | | | | | | |
| t (min) | 0 | 5 | 10 | 15 | 20 | 25 |
| **MF** | 100.00 | 101.94 ± 5.31### | 100.10 ± 5.38### | 100.29 ± 4.86### | 99.43 ± 6.47### | 98.57 ± 8.00### |
| | | 0.4316[a], 0.0004[b] | 0.9539, 0.0009 | 0.8530, 0.0002 | 0.7856, 0.0002 | 0.5851, 0.0003 |

Mean ± SD, $n = 10$ in each group.

** $p < 0.01$

*** $p < 0.001$ compared with the baseline (within a condition).

## $p < 0.01$

### $p < 0.001$ compared with the time-matched sham exposure (between conditions). $p$-value of sham in each column indicate the within difference. Two $p$-values of MF in each column indicate the within and between differences in this order. The Shapiro-Wilk test was performed to assess data distribution (Table A in S4 File). In the case of parametric distribution, $p$-values were calculated with the paired $t$-test (within a condition) and the Student's $t$-test (between conditions). In the case of nonparametric distribution, these were obtained with [a]the Wilcoxon signed-rank test (within a condition) and [b]the Wilcoxon rank-sum test (between conditions).

In the neck exposure, the repeated-measures ANOVA showed a significant Condition by Time interaction [$F(5, 90) = 11.82$, $p < 0.0001$]. No significant change in blood flow velocity values was induced by the MF exposure, but there were significant differences between the MF and sham exposures (Table 3D and Fig 3C).

## Effects of MF exposure on blood pressure and heart rate

The blood pressure (BP) and heart rate (HR) values were evaluated before (baseline) and after 15-min exposure to MFs at 10-min intervals in the forearm exposure. The systolic and diastolic BP and HR values (mean ± SD, $n = 10$) were measured at 10-min intervals of the pre-exposure, during the exposure (at 10 min after the start of the exposure), and the post-exposure (at 5 min after the termination of the exposure) (Table 4).

The repeated-measures ANOVA of BP showed no significant Condition by Time interaction [$F(2, 36) = 1.34$, $p = 0.2750$] and [$F(2, 36) = 1.78$, $p = 0.1826$] in systolic and diastolic BP, respectively. In MF exposure, the systolic BP values significantly decreased, but diastolic BP values did not significantly changed from the pre (baseline) values, and there was no significant difference between MF and sham exposures (Table 4). In addition, the repeated-measures ANOVA of HR showed no significant Condition by Time interaction [$F(2, 36) = 0.56$, $p = 0.5746$]. In MF and sham exposures, the HR values were not significantly changed from the pre (baseline) values and there was no significant difference between both conditions (Table 4).

**Table 4. Comparison of blood pressure (BP) and heart rate (HR) between 50 Hz MF and sham exposures in the forearm.**

| | Sham | | | MF | | |
|---|---|---|---|---|---|---|
| | **Pre** | **During** | **Post** | **Pre** | **During** | **Post** |
| Systolic BP (mmHg) | 116.8 ± 4.2 | 116.9 ± 3.3 | 115.7 ± 4.2 | 116.6 ± 4.3 | 113.8 ± 4.1 | 113.4 ± 3.4* |
| | – | 0.9246 | 0.1022 | – | 0.2059 | 0.0156[a] |
| | | | | 1.0000[b] | 0.0793 | 0.3803[b] |
| Diastolic BP (mmHg) | 72.7 ± 6.9 | 73.0 ± 9.9 | 73.4 ± 10.0 | 69.6 ± 5.1 | 73.4 ± 9.3 | 73.3 ± 10.0 |
| | – | 0.5020[a] | 1.0000[a] | – | 0.0521 0.9095[b] | 0.0547[a] |
| | | | | 0. 0691[b] | | 0.9095[b] |
| HR (bpm) | 63.9 ± 7.9 | 65.1 ± 7.2 | 65.6 ± 8.6 | 65.6 ± 7.3 | 64.0 ± 5.2 | 65.5 ± 5.5 |
| | – | 0.4074 | 0.2903 | – 0.6236 | 0.2851 0.7001 | 0.9720 0.9758 |

Mean ± SD, $n = 10$ in each group. $p$-value of sham in each column indicate the within difference. Two $p$-values of MF in each column indicate the within and between differences in this order. The Shapiro-Wilk test was performed to assess data distribution (Table A in S4 File). In the case of parametric distribution, $p$-values were calculated with the paired $t$-test (within a condition) and the Student's $t$-test (between conditions). In the case of nonparametric distribution, these were obtained with [a]the Wilcoxon signed-rank test (within a condition) and [b]the Wilcoxon rank-sum test (between conditions).

## Effects of MF exposure on hemoglobin oxygenation

The muscle hemodynamic changes of 10 male adults were monitored at two muscle regions of the left forearm twice on different days for MF and sham exposure experiments, and the mean hemoglobin oxygenation index (HOI) values were taken from CH-1 and CH-2 using fNIRS. The results showed that the HOI values in both MF and sham exposures in the forearm were peaked about 3–4 min after muscle loading exercise, indicating hyperperfusion (Fig 4A). This reflects the recovery time for hemoglobin/myoglobin desaturation in the capillary bed of exercising muscle [72]. When the values of 5-min exposure period were set at 100%, the recovery (decrease) rates of HOI values are shown in Table 5 and Fig 4B. The repeated-measures ANOVA of HOI showed no significant Condition by Time interaction [$F(4, 72) = 1.60$, $p = 0.1834$] in CH-1, and [$F(4, 72) = 1.02$, $p = 0.4042$] in CH-2. MF exposure showed faster and larger decrease rates of HOI values than those of sham exposure from the loading condition, but no significant difference was found between those of HOI values of MF and sham exposures at 5-min intervals (Table 5 and Fig 4B).

## Effects of MF exposure on ECG parameters

The ECG parameters were evaluated before (baseline) and after 15-min MF and sham exposures in the neck at 5-min intervals. The results of R-R interval (RRI) are shown in Table 6A and Fig 5A. The changing rate (%) from the baseline values of ECG parameters was also analyzed, primarily due to the large between-subjects variability. In sham exposure, the RRI values showed a tendency to be reduced during the motionless condition, which induced tachycardia, probably due to immobility-induced stress, and the RRI values were significantly decreased from the baseline value (indicated as 100%) (Table 6A and Fig 5A). In MF exposure, the RRI values were not significantly changed from the baseline value (Table 6A and Fig 5A). There were some significant differences between both conditions (Table 6A and Fig 5A). However, the repeated-measures ANOVA of RRI showed no significant Condition by Time interaction [$F(2, 36) = 0.88$, $p = 0.7092$].

The results of the high frequency (HF) and low-frequency/high frequency (LF/HF) ratio are shown in Table 6B and Fig 5B and Table 6C and Fig 5C, respectively. In the HF values, the repeated-measures ANOVA showed a significant Condition by Time interaction [$F(50, 900) = 1.67$, $p = 0.0030$]. In sham exposure, the HF values showed a tendency to be increased during

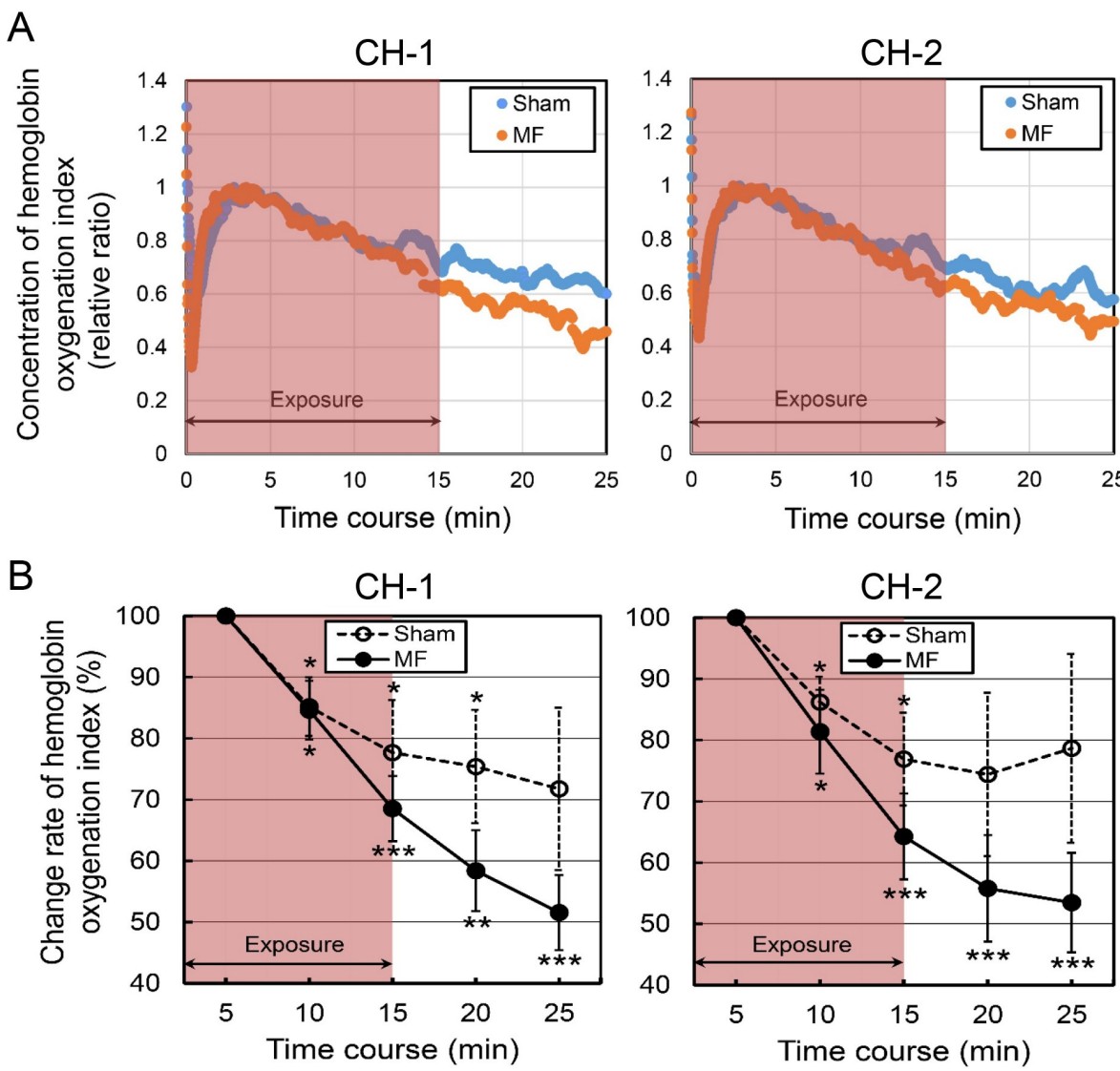

**Fig 4.** The time course of the mean concentration (A) and change rate (B) of hemoglobin oxygenation index values of flexor carpi radialis muscles in the forearm exposure to a 50 Hz MF after muscle loading exercise. CH-1: Middle part of flexor carpi radialis muscles. CH-2: Proximal part of flexor carpi radialis muscles. Relative peak values were set at 1.0 (A). The values of 5-min exposure period were set at 100% (B). The values are expressed as mean ± SEM. $n = 10$ in each condition. The duration of exposure is 15 min. MF exposure, red closed circles; sham exposure, blue open circles. Two-way repeated-measures ANOVA, followed by the Student's $t$-test or the Wilcoxon rank-sum test (between conditions), and the paired $t$-test or the Wilcoxon signed-rank test (within a condition). Values are expressed as mean ± SEM ($n = 10$ in each group). $^*p < 0.05$, $^{**}p < 0.01$, $^{***}p < 0.001$ compared with the baseline (within a group).

the motionless condition, and the HF values were significantly elevated from the baseline value at the 20-min condition. The MF exposure more significantly increased HF component values during the exposure and after the termination of the exposure periods from the baseline value and the HF value peaked at 14-min exposure from the baseline value by 74% (Table 6B and Fig 5B). With regard to the differences between the MF and sham exposures, the HF values were significantly increased by the MF exposure compared with the sham exposure intermittently during the exposure and after the termination of the exposure periods (Table 6B and Fig 5B). As the HF component is used as an index for parasympathetic activity [61], these results might be due to MF-induced increase in parasympathetic activity.

**Table 5. Comparison of change rate (%) of hemoglobin oxygenation index between 50 Hz MF and sham exposures in the forearm.**

**A. CH-1**

| $t$ (min) | 5 | 10 | 15 | 20 | 25 |
|---|---|---|---|---|---|
| **Sham** | 100.00 | 85.18 ± 5.05* | 77.67 ± 9.06* | 75.40 ± 9.77* | 71.75 ± 13.97 |
| | | 0.0166 | 0.0360 | 0.0329 | 0.0738 |
| **MF** | 100.00 | 84.62 ± 5.07* | 68.52 ± 5.60*** | 58.39 ± 6.98** | 51.53 ± 6.47*** |
| | | 0.0142, 0.9378 | 0.0003, 0.4039 | 0.0039[a], 0.2413[b] | 0.0000, 0.2124 |

**B. CH-2**

| $t$ (min) | 5 | 10 | 15 | 20 | 25 |
|---|---|---|---|---|---|
| **Sham** | 100.00 | 86.20 ± 4.41* | 76.92 ± 7.99* | 74.41 ± 14.07 | 78.65 ± 16.25 |
| | | 0.0121 | 0.0371[a] | 0.0840[a] | 0.3223[a] |
| **MF** | 100.00 | 81.38 ± 7.20* | 64.26 ± 7.41*** | 55.78 ± 9.17*** | 53.44 ± 8.57*** |
| | | 0.0295, 0.5769 | 0.0009, 0.3258[b] | 0.0009, 0.1306[b] | 0.0004, 0.2899[b] |

Mean ± SEM, $n = 10$ in each group.

*$p < 0.05$

**$p < 0.01$

***$p < 0.001$ compared with the baseline (within a condition). $p$-value of sham in each column indicate the within difference. Two $p$-values of MF in each column indicate the within and between differences in this order. The Shapiro-Wilk test was performed to assess data distribution (Table A in S4 File). In the case of parametric distribution, $p$-values were calculated with the paired $t$-test (within a condition) and the Student's $t$-test (between conditions). In the case of nonparametric distribution, these were obtained with [a]the Wilcoxon signed-rank test (within a condition) and [b]the Wilcoxon rank-sum test (between conditions).

In MF and sham exposures, the LF/HF ratio values showed a tendency to be increased from the baseline values under both conditions, probably due to immobility-induced stress, and the LF/HF ratio values were significantly elevated from the baseline value at the 5.5-min condition in sham exposure (Table 6C and Fig 5C). However, the repeated-measures ANOVA showed no significant Condition by Time interaction [$F(50, 900) = 0.67$, $p = 0.9608$]. No significant change in LF/HF ratio values was induced by the MF exposure, and there was no significant difference between the MF and sham exposures (Table 6C and Fig 5C).

## Effects of MF exposure on FMD

Brachial artery FMD is an indicator of vascular endothelial function and shear stress-induced nitric oxide (NO) production [62–64]. Therefore, MF-induced increase in FMD values largely indicates enhancement of NO production and NO-mediated vasodilator response. The FMD values were evaluated before (baseline) and after 30-min exposure to MFs as described previously [51]. The CV values for the baseline FMD values in the same individual were within 30% across the four testing session: 15.1% ± 9.1% (mean ± SD), range 4.1–29.9%, $n = 16$. These CV values are within the low range of published values for FMD (30%) in healthy individuals [64]. The repeated-measures ANOVA showed a significant Condition by Time interaction [$F(1, 30) = 11.98$, $p = 0.0016$]. The FMD values were significantly increased from the baseline value in the presence of MF exposure (Table 7 and Fig 6). There were significant differences between the MF and sham exposures (Fig 6). In contrast, in sham exposure, the FMD values showed a downward trend, partially due to decrease in endothelial NO synthase expression [73] but there was no significant change during the experimental period of about 30 min (Table 7 and Fig 6). As for this reason for physiological stability, in the case of the supine position without any intense intervention, it has been reported that the values of the forearm blood flow velocity, shear stress and vascular resistance were not changed for at least 5 h [74].

**Table 6. Comparison of change rate (%) of ECG parameters between 50 Hz MF and sham exposures in the neck.**

**A. RRI**

| $t$ (min) | 0 | 0.5 | 1 | 1.5 | 2 |
|---|---|---|---|---|---|
| Sham | 100.00 | 98.87 ± 5.59 | 100.49 ± 4.81 | 99.24 ± 4.21 | 98.31 ± 5.09 |
|  |  | 0.5387 | 0.7570 | 0.5825 | 0.3202 |
| MF | 100.00 | 99.87 ± 2.42 | 101.18 ± 6.58 | 102.72 ± 6.19 | 101.74 ± 7.77 |
|  |  | 0.8720, 0.6116 | 0.5852, 0.7915 | 0.1602[a], 0.3022[b] | 0.4977, 0.2606 |
| $t$ (min) | 2.5 | 3 | 3.5 | 4 | 4.5 |
| Sham | 97.35 ± 4.80 | 99.11 ± 5.93 | 99.47 ± 3.77 | 98.20 ± 5.12 | 98.75 ± 6.36 |
|  | 0.1143 | 0.6479 | 0.6672 | 0.2953 | 0.5496 |
| MF | 99.41 ± 5.37 | 100.78 ± 5.53 | 101.14 ± 8.83 | 101.74 ± 9.41 | 100.42 ± 7.86 |
|  | 0.7365, 0.3767 | 1.0000[a], 0.8199[b] | 0.6931, 0.5926 | 0.5724, 0.3135 | 0.8710, 0.6090 |
| $t$ (min) | 5 | 5.5 | 6 | 6.5 | 7 |
| Sham | 98.65 ± 5.03 | 96.77 ± 5.77 | 97.55 ± 5.77 | 97.20 ± 6.16 | 98.60 ± 6.24 |
|  | 0.4169 | 0.1107 | 0.2131 | 0.1844 | 0.4967 |
| MF | 99.74 ± 6.94 | 100.18 ± 7.40 | 100.46 ± 8.24 | 99.99 ± 7.28 | 100.27 ± 7.07 |
|  | 0.9066, 0.6932 | 0.9411, 0.2671 | 0.8629, 0.3740 | 0.9982, 0.3665 | 0.9055, 0.5824 |
| $t$ (min) | 7.5 | 8 | 8.5 | 9 | 9.5 |
| Sham | 99.86 ± 5.79 | 98.12 ± 5.31 | 98.98 ± 5.32 | 97.51 ± 3.67 | 97.66± 4.68 |
|  | 0.9413 | 0.2921 | 0.5588 | 0.0840[a] | 0.7552 |
| MF | 100.13 ± 6.01 | 101.10 ± 5.76 | 99.39 ± 6.51 | 100.64 ± 8.16 | 101.25 ± 7.64 |
|  | 0.9474, 0.9204 | 0.5607, 0.2447 | 0.7729, 0.8796 | 0.8098, 0.4950[b] | 0.6161, 0.2242 |
| $t$ (min) | 10 | 10.5 | 11 | 11.5 | 12 |
| Sham | 98.14 ± 4.97 | 97.83 ± 3.27 | 97.68 ± 3.60 | 95.76 ± 3.16** | 96.42 ± 2.51** |
|  | 0.2667 | 0.0652 | 0.0725 | 0.0022 | 0.0015 |
| MF | 101.48 ± 7.44 | 100.60 ± 6.76 | 101.84 ± 7.69 | 102.18 ± 7.60# | 101.31 ± 6.58# |
|  | 0.5451, 0.2554 | 0.7851, 0.2640 | 0.4687, 0.1461 | 0.3887, 0.0297 | 0.5455, 0.0494 |
| $t$ (min) | 12.5 | 13 | 13.5 | 14 | 14.5 |
| Sham | 97.74 ± 3.41 | 98.23 ± 3.80 | 97.20 ± 3.87* | 96.97 ± 3.49* | 95.04 ± 4.51** |
|  | 0.0655 | 0.1742 | 0.0478 | 0.0227 | 0.007 |
| MF | 100.65 ± 6.23 | 99.41 ± 5.53 | 98.62 ± 4.30 | 100.16 ± 3.24# | 100.06 ± 7.88 |
|  | 0.7507, 0.2169 | 0.7449, 0.5834 | 0.3350, 0.4486 | 0.8757, 0.0481 | 0.9804, 0.1019 |
| $t$ (min) | 15 | 15.5 | 16 | 16.5 | 17 |
| Sham | 97.72 ± 4.45 | 97.64 ± 4.74 | 98.11 ± 3.92 | 96.86 ± 5.07 | 95.99 ± 5.76 |
|  | 0.1403 | 0.1496 | 0.1615 | 0.0823 | 0.0552 |
| MF | 98.47 ± 8.87 | 99.37 ± 8.98 | 98.32 ± 9.59 | 100.06 ± 7.28 | 97.39 ± 9.49 |
|  | 0.5991, 0.8148 | 0.8293, 0.5986 | 0.6250[a], 0.4249[b] | 0.9798, 0.2711 | 0.4070, 0.6962 |
| $t$ (min) | 17.5 | 18 | 18.5 | 19 | 19.5 |
| Sham | 96.27 ± 4.96* | 95.60 ± 4.33* | 94.48 ± 3.86** | 95.71 ± 5.44* | 95.1 ± 2.79*** |
|  | 0.0415 | 0.0106 | 0.0014 | 0.0117[a] | 0.0004 |
| MF | 99.19 ± 10.30 | 99.23 ± 10.19 | 99.64 ± 6.82 | 101.48 ± 7.00# | 98.70 ± 10.89 |
|  | 0.8086, 0.4345 | 0.8165, 0.3197 | 0.8694, 0.0559 | 0.5207, 0.0443[b] | 0.7153, 0.3346 |
| $t$ (min) | 20 | 20.5 | 21 | 21.5 | 22 |
| Sham | 97.88 ± 3.74 | 97.14 ± 5.61 | 96.87 ± 5.79 | 96.87 ± 5.93 | 96.74 ± 4.42* |
|  | 0.0840[a] | 0.1418 | 0.1211 | 0.1602[a] | 0.0446 |
| MF | 98.97 ± 10.24 | 99.63 ± 7.78 | 100.41 ± 5.08 | 99.85 ± 4.89 | 98.27 ± 7.20 |
|  | 0.7581, 0.5696[b] | 0.8822, 0.4245 | 0.8063, 0.1638 | 0.9234, 0.1370[b] | 0.4654, 0.5768 |
| $t$ (min) | 22.5 | 23 | 23.5 | 24 | 24.5 |

*(Continued)*

**Table 6.** (Continued)

| Sham | 97.38 ± 3.96 | 97.79 ± 5.41 | 99.09 ± 5.54 | 97.77 ± 5.32 | 98.3 ± 4.82 |
|---|---|---|---|---|---|
| | 0.0657 | 0.2286 | 0.6176 | 0.2178 | 0.2923 |
| MF | 99.20 ± 7.15 | 97.47 ± 7.45 | 97.26 ± 9.88 | 100.15 ± 7.79 | 99.38 ± 7.57 |
| | 0.7311, 0.4926 | 0.3115, 0.9146 | 0.4031, 0.6164 | 0.9540, 0.4378 | 0.8001, 0.7088 |
| $t$ (min) | 25 | | | | |
| Sham | 98.68 ± 3.99 | | | | |
| | 0.3208 | | | | |
| MF | 100.6 ± 8.88 | | | | |
| | 0.8345, 0.5422 | | | | |

**B. HF**

| $t$ (min) | 0 | 0.5 | 1 | 1.5 | 2 |
|---|---|---|---|---|---|
| Sham | 100.00 | 110.52 ± 36.41 | 97.9 ± 38.80 | 91.07 ± 35.23 | 88.76 ± 32.37 |
| | | 0.3845 | 0.8679 | 0.4433 | 0.3008 |
| MF | 100.00 | 104.5 ± 24.93 | 102.26 ± 51.65 | 116.09 ± 56.81 | 116.36 ± 53.53 |
| | | 0.5818, 0.6720 | 0.8929, 0.8335 | 0.6953[a], 0.4272[b] | 0.3591, 0.1836 |
| $t$ (min) | 2.5 | 3 | 3.5 | 4 | 4.5 |
| Sham | 92.54 ± 36.61 | 93.85 ± 37.03 | 90.41 ± 42.80 | 102.04 ± 32.39 | 98.32 ± 22.56 |
| | 0.5352 | 0.6121 | 0.4966 | 0.8466 | 0.8189 |
| MF | 129.93 ± 53.14 | 120.12 ± 57.59 | 114.59 ± 39.87 | 123.03 ± 53.90 | 100.97 ± 40.71 |
| | 0.1086, 0.0856 | 0.6250[a], 0.3845[b] | 0.2771, 0.2078 | 0.2754[a], 0.4494[b] | 0.9418, 0.8598 |
| $t$ (min) | 5 | 5.5 | 6 | 6.5 | 7 |
| Sham | 98.73 ± 38.31 | 109.2 ± 35.22 | 104.53 ± 37.31 | 107.07 ± 46.94 | 112.58 ± 40.31 |
| | 0.9187 | 0.4299 | 0.7099 | 0.6452 | 0.3494 |
| MF | 116.27 ± 33.43 | 126.71 ± 31.82* | 132.1 ± 65.61 | 148.47 ± 86.14* | 145.3 ± 56.77* |
| | 0.1582, 0.2900 | 0.0263, 0.2590 | 0.1562, 0.2670 | 0.0273[a], 0.2567[b] | 0.0326, 0.1565 |
| $t$ (min) | 7.5 | 8 | 8.5 | 9 | 9.5 |
| Sham | 117.48 ± 54.29 | 107.36 ± 45.61 | 112.04 ± 58.07 | 106.24 ± 51.73 | 106.13 ± 48.7 |
| | 0.3353 | 0.6222 | 0.5285 | 0.7119 | 0.2168 |
| MF | 133.25 ± 41.7* | 129.48 ± 66.61 | 131.26 ± 53.08 | 172.37 ± 62.61**, # | 146.92 ± 76.21* |
| | 0.0327, 0.4763 | 0.1952, 0.3991 | 0.0955, 0.4499 | 0.0053, 0.0194 | 0.0488[a], 0.2729[b] |
| $t$ (min) | 10 | 10.5 | 11 | 11.5 | 12 |
| Sham | 117.17 ± 63.93 | 114.75 ± 51.58 | 136.63 ± 55.42 | 135.45 ± 56.34 | 119.44 ± 44.11 |
| | 0.4176 | 0.3895 | 0.0662 | 0.0778 | 0.1968 |
| MF | 135.46 ± 62.91 | 148.39 ± 40.08** | 127.50 ± 43.79 | 132.72 ± 55.23 | 136.01 ± 92.72 |
| | 0.1084, 0.5273 | 0.0098[a], 0.0756[b] | 0.0783, 0.6879 | 0.0938, 0.9140 | 0.3223[a], 1.0000[b] |
| $t$ (min) | 12.5 | 13 | 13.5 | 14 | 14.5 |
| Sham | 123.67 ± 52.72 | 107.39 ± 56.8 | 110.71 ± 62.49 | 116.47 ± 45.93 | 102.38 ± 50.60 |
| | 0.1893 | 0.6905 | 0.6011 | 0.2862 | 0.8849 |
| MF | 160.54 ± 89.08 | 157.64 ± 85.34* | 150.13 ± 87.66 | 174.29 ± 89.01* | 167.28 ± 63.38**, # |
| | 0.0601, 0.2781 | 0.0273[a], 0.1041[b] | 0.1309[a], 0.2121[b] | 0.0269, 0.0901 | 0.0084, 0.0214 |
| $t$ (min) | 15 | 15.5 | 16 | 16.5 | 17 |
| Sham | 100.09 ± 36.42 | 94.61 ± 43.15 | 116.89 ± 59.02 | 116.63 ± 60.12 | 120.32 ± 54.17 |
| | 0.9941 | 0.7022 | 0.3891 | 0.4045 | 0.2659 |
| MF | 166.24 ± 59.09**, ## | 159.97 ± 52.25**, ## | 140.28 ± 43.86* | 135.14 ± 63.04 | 135.37 ± 67.91 |
| | 0.0020[a], 0.0040[b] | 0.0055, 0.0071 | 0.0175, 0.3288 | 0.1118, 0.5101 | 0.1309[a], 0.7334[b] |
| $t$ (min) | 17.5 | 18 | 18.5 | 19 | 19.5 |
| Sham | 125.74 ± 43.5 | 113.04 ± 48.57 | 124.81 ± 62.44 | 141.37 ± 76.53 | 129.36 ± 49.54 |
| | 0.0941 | 0.4177 | 0.2406 | 0.1216 | 0.0936 |

*(Continued)*

**Table 6.** (Continued)

| MF | 160.09 ± 67.11* | 153.27 ± 58.26* | 129.88 ± 48.32 | 126.18 ± 62.91 | 145.09 ± 75.72 |
|---|---|---|---|---|---|
| | 0.0197, 0.1940 | 0.0178, 0.1114 | 0.0822, 0.8415 | 0.2207, 0.6340 | 0.0923, 0.5904 |
| $t$ (min) | 20 | 20.5 | 21 | 21.5 | 22 |
| Sham | 148.32 ± 50.79* | 142.36 ± 60.47 | 148.01 ± 71.95 | 137.55 ± 58.65 | 137.56 ± 76.64 |
| | 0.0147 | 0.0540 | 0.0641 | 0.0736 | 0.1556 |
| MF | 132.74 ± 38.68* | 148.62 ± 67.81* | 139.64 ± 74.33 | 133.27 ± 66.76 | 144.03 ± 75.64 |
| | 0.0254, 0.4509 | 0.0496, 0.8300 | 0.1260, 0.8010 | 0.1495, 0.8807 | 0.0988, 0.8514 |
| $t$ (min) | 22.5 | 23 | 23.5 | 24 | 24.5 |
| Sham | 172.81 ± 81.91* | 163.78 ± 83.11* | 134.98 ± 87.29 | 141.7 ± 75.05 | 140.39 ± 81.78 |
| | 0.0203 | 0.0382 | 0.2369 | 0.1128 | 0.1528 |
| MF | 158.27 ± 88.52 | 143.67 ± 62.14 | 131.08 ± 52.60* | 127.95 ± 50.07 | 122.69 ± 50.84 |
| | 0.0671, 0.7076 | 0.0534, 0.5482 | 0.0371[a], 0.9698[b] | 0.1113, 0.6365 | 0.1917, 0.5698 |
| $t$ (min) | 25 | | | | |
| Sham | 124.46 ± 79.50 | | | | |
| | 0.3560 | | | | |
| MF | 117.78 ± 36.43 | | | | |
| | 0.1934[a], 0.7313[b] | | | | |
| **C. LF/HF** | | | | | |
| $t$ (min) | 0 | 0.5 | 1 | 1.5 | 2 |
| Sham | 100.00 | 108.97 ± 41.92 | 103.34 ± 18.80 | 95.54 ± 31.03 | 131.21 ± 50.58 |
| | | 0.5157 | 0.5879 | 0.3223[a] | 0.0828 |
| MF | 100.00 | 123.4 ± 71.61 | 147.07 ± 125.65 | 117.24 ± 72.5 | 133.4 ± 110.32 |
| | | 0.3284, 0.5908 | 0.8457[a], 0.6499[b] | 0.9219[a], 0.8202[b] | 0.5566[a], 0.5966[b] |
| $t$ (min) | 2.5 | 3 | 3.5 | 4 | 4.5 |
| Sham | 113.62 ± 53.08 | 103.1 ± 51.26 | 142.21 ± 85.00 | 140.43 ± 72.44 | 123.77 ± 60.37 |
| | 0.4379 | 0.8526 | 0.1508 | 0.1114 | 0.2446 |
| MF | 121.38 ± 80.36 | 118.86 ± 92.54 | 112.95 ± 58.22 | 111.20 ± 58.75 | 139.52 ± 84.36 |
| | 0.5566[a], 0.9397[b] | 0.9219[a], 0.9698[b] | 0.4997, 0.3824 | 0.5615, 0.3354 | 0.1726, 0.6375 |
| $t$ (min) | 5 | 5.5 | 6 | 6.5 | 7 |
| Sham | 136.99 ± 73.79 | 150.09 ± 59.84* | 126.85 ± 51.07 | 106.59 ± 49.79 | 113.52 ± 48.86 |
| | 0.1474 | 0.0266 | 0.1308 | 0.6852 | 0.4044 |
| MF | 119.31 ± 66.73 | 106.10 ± 62.85 | 118.64 ± 94.83 | 108.24 ± 77.35 | 103.61 ± 65.73 |
| | 0.3839, 0.5813 | 0.7657, 0.1264 | 1.0000[a], 0.4492[b] | 0.8457[a], 0.7335[b] | 1.0000[a], 0.3642[b] |
| $t$ (min) | 7.5 | 8 | 8.5 | 9 | 9.5 |
| Sham | 124.19 ± 61.48 | 134.6 ± 108.42 | 139.92 ± 68.05 | 149.65 ± 75.87 | 125.99 ± 59.12 |
| | 0.2448 | 0.3392[a] | 0.0966 | 0.0684 | 0.8579 |
| MF | 101.05 ± 40.29 | 101.69 ± 37.24 | 102.24 ± 45.32 | 124.42 ± 68.14 | 124.35 ± 80.94 |
| | 0.9359, 0.3347 | 0.4922[a], 1.0000[b] | 0.8793, 0.1648 | 0.2864, 0.4441 | 0.3663, 0.9594 |
| $t$ (min) | 10 | 10.5 | 11 | 11.5 | 12 |
| Sham | 130.79 ± 69.74 | 171.64 ± 118.47 | 166.73 ± 97.49 | 145.24 ± 78.01 | 163.07 ± 114.49 |
| | 0.1962 | 0.5566[a] | 0.0586 | 0.0534 | 0.1155 |
| MF | 127.68 ± 72.60 | 126.1 ± 98.03 | 117.63 ± 77.90 | 120.57 ± 92.61 | 163.02 ± 114.57 |
| | 0.2587, 0.9233 | 0.4216, 0.4922[b] | 0.4924, 0.2301 | 0.7695[a], 0.2897[b] | 0.1160, 0.9993 |
| $t$ (min) | 12.5 | 13 | 13.5 | 14 | 14.5 |
| Sham | 149.03 ± 101.02 | 95.32 ± 44.97 | 99.84 ± 40.42 | 108.16 ± 56.11 | 130.94 ± 89.58 |
| | 0.1592 | 0.7495 | 0.9905 | 0.6566 | 0.3031 |
| MF | 130.98 ± 127.59 | 101.78 ± 60.11 | 134.35 ± 57.12 | 101.33 ± 39.45 | 116.52 ± 94.55 |
| | 0.8457[a], 0.4495[b] | 1.0000[a], 0.6497[b] | 0.0896, 0.1382 | 0.9177, 0.7569 | 0.7695[a], 0.5964[b] |

(*Continued*)

**Table 6.** (Continued)

| t (min) | 15 | 15.5 | 16 | 16.5 | 17 |
|---|---|---|---|---|---|
| Sham | 126.41 ± 90.12 | 117.42 ± 40.47 | 115.4 ± 50.24 | 94.29 ± 34.12 | 125.64 ± 70.59 |
| | 0.1602[a] | 0.2065 | 0.6250[a] | 0.6097 | 0.3683[a] |
| MF | 106.08 ± 86.74 | 89.50 ± 48.48 | 104.22 ± 43.14 | 88.32 ± 31.65 | 106.82 ± 89.24 |
| | 0.4316[a], 0.4958[b] | 0.2324[a], 0.1301[b] | 0.7640, 0.6774[b] | 0.2731, 0.6895 | 1.0000[a], 0.0961[b] |
| t (min) | 17.5 | 18 | 18.5 | 19 | 19.5 |
| Sham | 117.74 ± 62.38 | 106.27 ± 54.44 | 101.84 ± 56.09 | 133.35 ± 51.03 | 142.58 ± 76.22 |
| | 0.3919 | 0.5566[a] | 0.5566[a] | 0.0687 | 0.1111 |
| MF | 119.63 ± 73.25 | 91.20 ± 54.92 | 92.60 ± 50.36 | 102.49 ± 68.52 | 115.60 ± 65.78 |
| | 0.4186, 0.9510 | 0.1602[a], 0.5703[b] | 0.6531, 0.7051[b] | 0.7695[a], 0.1123[b] | 0.4725, 0.4081 |
| t (min) | 20 | 20.5 | 21 | 21.5 | 22 |
| Sham | 124.43 ± 89.27 | 128.04 ± 66.16 | 117.18 ± 59.2 | 108.55 ± 51.34 | 109.82 ± 56.06 |
| | 0.4094 | 0.2131 | 0.3829 | 0.6110 | 0.5931 |
| MF | 120.33 ± 79.21 | 109.64 ± 48.70 | 93.66 ± 51.62 | 91.75 ± 43.29 | 100.81 ± 50.54 |
| | 0.4380, 0.9147 | 0.5470, 0.4887 | 0.7068, 0.3566 | 0.5617, 0.4394 | 0.9605, 0.7104 |
| t (min) | 22.5 | 23 | 23.5 | 24 | 24.5 |
| Sham | 126.28 ± 66.46 | 124.59 ± 70.27 | 125.91 ± 81.91 | 111.11 ± 65.23 | 119.63 ± 59.93 |
| | 0.2428 | 0.2972 | 0.3434 | 0.6030 | 0.3273 |
| MF | 108.40 ± 86.61 | 116.44 ± 89.30 | 127.30 ± 77.87 | 118.10 ± 84.22 | 114.28 ± 61.61 |
| | 0.6250[a], 0.2410[b] | 0.5748, 0.8233 | 0.2963, 0.9693 | 0.5138, 0.8382 | 0.4822, 0.8462 |
| t (min) | 25 | | | | |
| Sham | 139.32 ± 65.17 | | | | |
| | 0.0840[a] | | | | |
| MF | 108.13 ± 48.40 | | | | |
| | 0.6081, 0.4055[b] | | | | |

Mean ± SD, $n = 10$ in each group.

[*]$p < 0.05$

[**]$p < 0.01$

[***]$p < 0.001$ compared with the baseline (within a condition).

[#]$p < 0.05$

[##]$p < 0.01$ compared with the time-matched sham exposure (between conditions). *p*-value of sham in each column indicate the within difference. Two *p*-values of MF in each column indicate the within and between differences in this order. The Shapiro-Wilk test was performed to assess data distribution (Table A in S4 File). In the case of parametric distribution, *p*-values were calculated with the paired *t*-test (within a condition) and the Student's *t*-test (between conditions). In the case of nonparametric distribution, these were obtained with [a]the Wilcoxon signed-rank test (within a condition) and [b]the Wilcoxon rank-sum test (between conditions).

## Dosimetry of electric fields (EFs) induced by a 50 Hz MF

Fig 7 shows the estimated distribution of the induced EFs for the three exposure conditions. In terms of the EF intensity at the body surface, the simulation results indicated that the highest EF strengths were located in the regions underneath the coils, with minor diffusion to other regions. The results also indicated a strong effect of the body-coil distance on the induced EF strength, which explains the relative differences in the EF strength between the two coils (Fig 7A and 7B). Since the coil positioning is expected to differ between the computer simulations and the actual subjects, the calculated EFs should therefore be treated as order-of -magnitude estimates.

Table 8 lists the estimated maximum values of the induced EF strengths in various tissues. Computer simulations of the induced EFs indicated that the order-of-magnitude estimates of the peak values were 100–500 mV/m, depending on the exposure conditions. The results

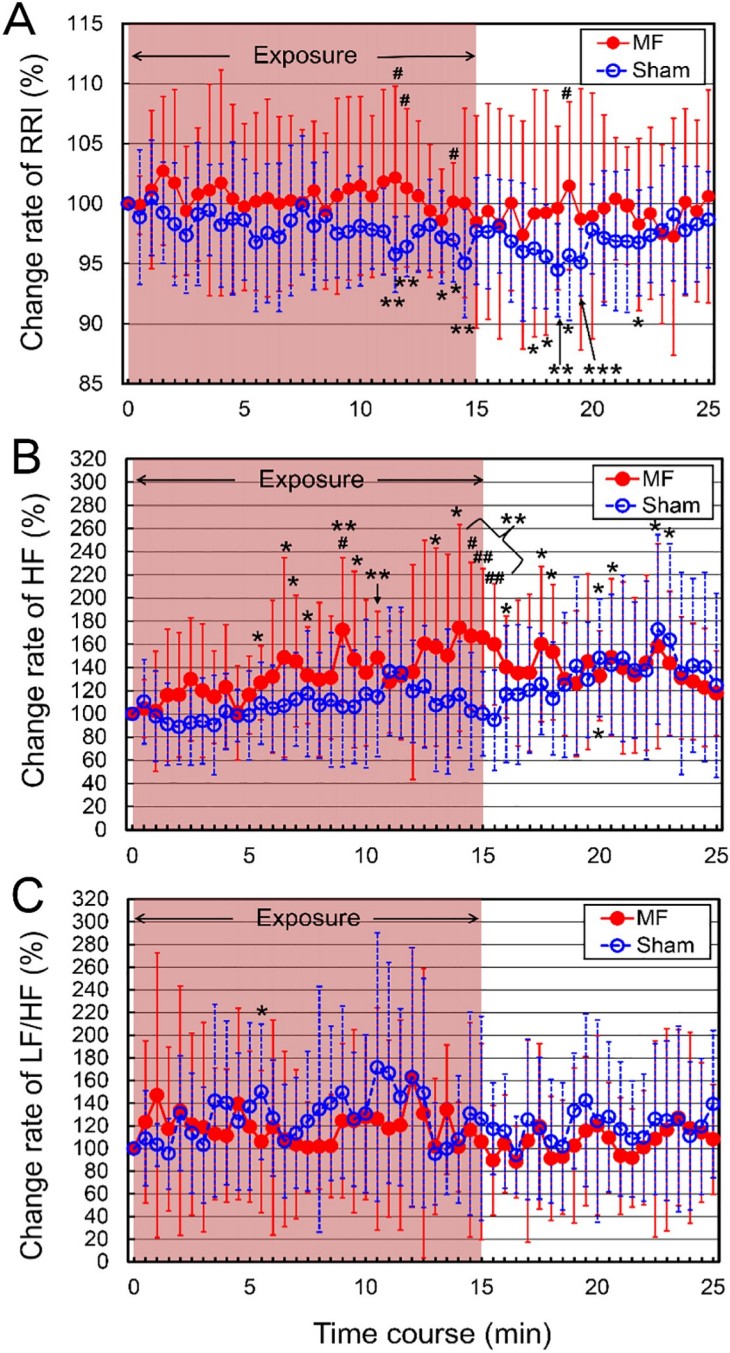

**Fig 5. The time course of change rate (%) in ECG parameters in the neck exposure to a 50 Hz MF.** (A) R-R interval (RRI) of heart rate variability. (B) High-frequency (HF) component. (C) Low-frequency/high-frequency (LF/HF) ratio. The values are expressed as mean ± SD. $n = 10$ in each condition. The duration of exposure is 15 min. MF exposure, red closed circles; sham exposure, blue open circles. Two-way repeated-measures ANOVA, followed by the Student's *t*-test or the Wilcoxon rank-sum test (between conditions), and the paired *t*-test or the Wilcoxon signed-rank test (within a condition). *$p < 0.05$, **$p < 0.01$, ***$p < 0.001$ compared with the baseline (within a condition). #$p < 0.05$, ##$p < 0.01$ compared with the time-matched sham exposure (between conditions).

**Table 7. Comparison of FMD (%) between 50 Hz MF and sham exposures in the upper arm.**

| $t$ (min) | 0 | 30 |
|---|---|---|
| **Sham** | 8.11 ± 1.73 | 7.41 ± 1.46 |
| | – | 0.0613 |
| **MF** | 7.89 ± 2.09 | 9.00 ± 1.58[*, ##] |
| | – | 0.0126 |
| | 0.7425 | 0.0060 |

Mean ± SD, $n$ = 16 in each group.

[*]$p < 0.05$ compared with the baseline (within a condition).

[##]$p < 0.01$ compared with the time-matched sham exposure (between conditions). $p$-value of sham in each column indicate the within difference. Two $p$-values of MF in each column indicate the within and between differences in this order. Parametric distributions in all cases were confirmed with the Shapiro-Wilk test (Table A in S4 File), and thereby, $p$-values were calculated with the paired $t$-test (within a condition) and the Student's $t$-test (between conditions).

indicated that the induced EF strengths were considerably weaker for the neck exposure compared to the forearm and upper arm exposures. For all three exposure conditions, the highest EF strengths were found in the skin and subcutaneous fat. The induced EFs in blood vessels, including large arteries and veins, were maximal in the forearm exposure condition.

## Discussion

The present study examined the acute effects of a 50 Hz MF on the values of blood flow velocity, blood pressure, heart rate, blood oxygenation, ECG parameters and FMD in healthy young men. Experiments were carried out in a prospective, randomized, double-blind, sham-controlled, counterbalanced, crossover manner using an MF exposure device. Participants were regionally exposed to the MF at $B_{max}$ 180 mT or $B_{rms}$ 127 mT for up to 30 min. The results of the blood flow velocity as the primary outcome measure showed that regional MF exposures significantly increased blood flow velocity compared to sham exposure (Fig 3). After the termination of the MF exposure, the MF effects on the blood flow velocity were sustained for at least 10 min. Considering the prolonged immobility conditions in subjects, we have not examined the prolonged post-exposure effects. When subjects leave the left forearm still on the exposure device for more than 30 min, the blood flow velocity will remain stagnant like a pressure ulcer. Here, the mean external pressure levels against the dorsal side of the forearm resulting from body weight support were 56.7 mmHg ± 4.1 mmHg (mean ± SD, $n$ = 10), which were measured using a portable interface pressure sensor (Palm Q, CAPE, Yokosuka, Japan). The interface pressure level of 60 mmHg applied for 1 h or more is considered to impede blood flow and induce tissue damage [69]. More recently, the capillary pressures excessively higher than 32 mmHg for 2 h or more are considered to cause tissue damage, and therefore, the pressure of 32 mmHg is regarded as the capillary closing pressure threshold [70].

At 10 min after the termination of regional exposure, the mean values of blood flow velocity in MF exposure were still higher than those in sham exposure by about 12%, 11% and 14% under forearm, upper arm and neck exposures, respectively (Fig 3). In these cases, the mean decrease rate of blood flow velocity in the post-exposure period was about 1.5%/min for MF exposure and about 0.5%/min for sham exposure. Therefore, it is estimated that the MF effects will last for at least 20-min post-exposure. In another way, however, when subjects lift the forearm away from the device and move it freely, the blood flow velocity will recover to the baseline level in a few minutes, and the significant differences in the blood flow velocity between

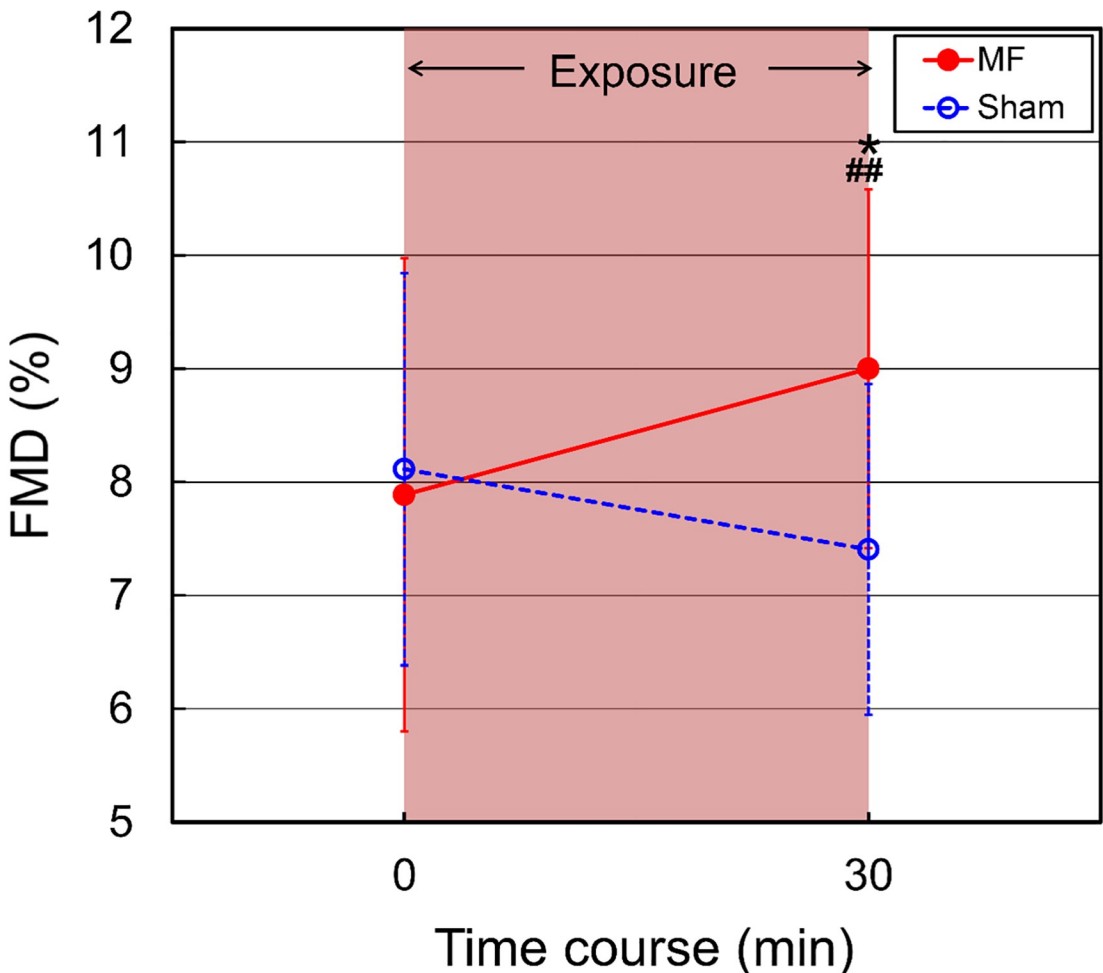

**Fig 6. The time course of changes in FMD in the upper arm exposure to a 50 Hz MF.** The duration of exposure is 30 min. MF exposure, red closed circles; sham exposure, blue open circles. Values are expressed as mean ± SD. $n$ = 16 in each condition. Two-way repeated-measures ANOVA, followed by the Student's $t$-test or the Wilcoxon rank-sum test (between conditions), and the paired $t$-test or the Wilcoxon signed-rank test (within a condition). *$p < 0.05$ compared with the baseline (within a condition). ##$p < 0.01$ compared with the time-matched sham exposure (between conditions).

both exposures will disappear. Therefore, the prolonged post-exposure effects remain elusive and further research is necessary to clarify the effects.

The increased effects on the blood flow velocity by the MF exposure (50 Hz, 1 mT, for 10 min) were consistent with those of the previous animal study under anesthesia [32]. To clarify these neuroelectric mechanisms of time-varying MFs, the relationship between nerve stimulation and increased blood circulation has been examined mainly using low-frequency repetitive transcranial magnetic stimulation (rTMS) [75, 76]. In contrast, however, the decreases in blood circulation induced by rTMS have been reported [76, 77]. Thus, the effects of rTMS on blood circulation are variable and fluctuating, depending on the stimulus regions, conditions and/or configurations, e.g., stimulus frequency, intensity, duration and repeated sessions, have not been established, and moreover, as is the case of 50 Hz MF effects examined in this study, the prolonged post-exposure effects also have not been clarified.

The results of fNIRS experiments showed that hemoglobin oxygenation index (HOI) values in both MF and sham exposures were peaked about 3–4 min after muscle loading exercise (Fig

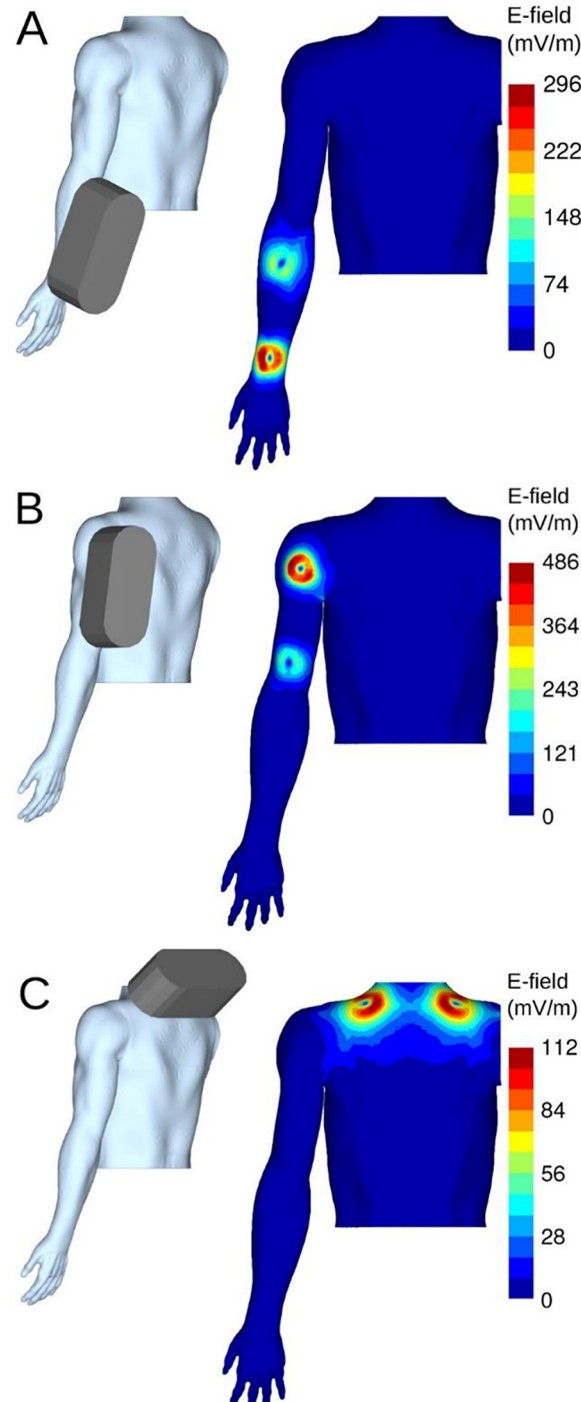

**Fig 7. Modeled distribution of the induced EFs in three different regional exposures to a 50 Hz MF.** (A) Forearm exposure. (B) Upper arm exposure. (C) Neck exposure. The strength of the induced EFs is visualized at the depth of 2 mm below the skin surface.

4A). MF exposure recovered HOI values faster and higher than sham exposure from the loading condition, but no significant difference was found between those of HOI values of MF and sham exposures (Fig 4B). These findings suggest that MF exposure could partially improve

**Table 8. The peak EF strengths (mV/m) in tissues induced by a 50 Hz MF for the forearm, upper arm and neck exposures.**

| Tissue | Forearm | Upper arm | Neck |
|---|---|---|---|
| Skin | 319.3 | 491.7 | 122.0 |
| Fat | 282.3 | 453.9 | 96.9 |
| Muscle | 141.9 | 382.8 | 63.5 |
| Blood vessel | 142.8 | 39.0 | 23.5 |
| Bone | 237.8 | 98.9 | 82.3 |
| Heart | 1.1 | 4.4 | 11.4 |

autonomic responses to maintain the homeostatic balance in the recovery process in increased circulation after exercise. This is because MF can actively enhance a better blood recovery compared to the normal recovery process, indicating faster hemodynamic adaptation to inflammatory conditions.

The response of electrically excitable nerve and muscle tissues to induced EFs has been demonstrated [33–35]. The EFs stimulate the tissues in the exposed region when the exposure condition exceeds the threshold of tissues. The most robustly established effect of EFs below the threshold for direct nerve or muscle excitation is the induction of magnetophosphenes, which is the perception of faint flickering light in the periphery of the visual field [35]. In this case of retina, the threshold for induced EF intensity has been estimated to lie between about 50 and 100 mV/m at 20 Hz, rising at higher and lower frequencies. In the present experiment at 50 Hz, however, we have not detected the magnetophosphene perception by any participants. In vitro experimental studies using brain slices suggest that minimum thresholds for nerve cell activity lie below frequencies of 100 Hz and may be as low as 100 mV/m [34].

Except for retina, brain and heart, however, the induced EFs in tissues have not yet been estimated and discussed in the field of 50 Hz MF, especially, at the high intensity such as 180 mT. The actual induced EFs depend strongly on the subject, e.g., the individual anatomy, posture and coil position. In addition, electrical conductive inhomogeneity and anisotropy of tissues, such as myocardium [37] and different tissue conductivities [38] strongly guide the resulting variability in induced EFs. Therefore, in this study, computer simulations were performed as order-of-magnitude estimates. The simulations indicated that the order-of-magnitude estimates of the peak induced EFs were 100–500 mV/m, depending on the exposure conditions (Table 8). The induced EFs were highest underneath the coils, with minor diffusion to other regions. The peak values of induced EFs in blood vessels were 143 mV/m in the forearm exposure, 39 mV/m in the upper arm exposure, and 24 mV/m in the neck exposure (Table 8). The increase rate of ulnar artery blood flow velocity being exposed in three different regions (Fig 3) is likely dependent on and positively correlated with the induced EFs in blood vessels. However, the induced EF strength is below the basic restrictions of ICNIRP (0.8 V/m in all tissues of head and body) [35], or the dosimetric reference limit of IEEE-ICES (2.1 V/m in hands, wrists, feet and ankles) [36]. Hence, it seems likely that the EF strength are far too weak to cause direct myelinated nerve stimulation, or at least could not induce adverse effects in terms of rheobase excitation thresholds. On the other hand, the peak MF strength in the case of 50 Hz are above the basic restrictions of IEEE-ICES ($B_{max}$ 107 mT or $B_{rms}$ 76 mT in arms or legs and $B_{max}$ 3.83 mT or $B_{rms}$ 2.71 mT in torso) [36].

We speculate that regional MF strength could play a crucial role in the initiation of a series of the physiological response processes involved in hemodynamic responses. Our results demonstrated that ulnar artery blood flow velocity values were significantly increased by the regional MF exposures compared with the sham exposures. The estimated $B_{max}$ ($B_{rms}$) values

in an ulnar artery (forearm), a brachial artery (upper arm), and a carotid artery (neck) are approximately 13 mT (9.2 mT), 8 mT (5.7 mT), and 5 mT (3.5 mT) respectively (Fig F in S4 File), and the blood flow velocity increased corresponding to the $B_{max}$ values (Fig 3), which are also apparently positively correlated with the induced EFs in blood vessels (Fig 7). Thus, these MF effects seem to be a field strength-dependent and may be region-specific.

The diastolic BP and HR values were not significantly affected by the regional MF exposure (Table 4). Therefore, it seems plausible that this regional MF exposure method did not strongly affect systemic hemodynamics or whole body circulatory response in healthy young men. Interestingly, the results of ECG parameters showed that parasympathetic HF activity was significantly increased by the neck MF exposure compared with the sham exposure intermittently during the exposure and after the termination of the exposure periods. In contrast, no significant change in LF/HF ratio values was induced by the MF exposure. These results suggest that the MF exposure can activate parasympathetic activity and thereby lead to increase vasodilation and blood flow. In addition, from the results of FMD test, the MF exposure significantly increased FMD values compared to the baseline value as well as compared to the sham exposure. The results were consistent with our previous findings investigated with smaller sample sizes [51]. As FMD is an indicator of vascular endothelial function and shear stress-induced nitric oxide (NO) production [62–64], the MF-induced increase of FMD values is largely related to enhancement of NO production [51]. These findings suggest that this 50 Hz MF exposure can activate parasympathetic nerves and induce hemodynamic responses via cholinergic mediators together with NO-mediated vasodilation.

The hemodynamic mechanisms of 50 Hz MFs have been reviewed [27, 30, 31], and we have proposed the mechanisms by which 50 Hz MFs could alter acetylcholine (ACh) and NO signaling pathways [50, 51] based on the following reports [5,39,40,41,42]. ACh has been found to be released from some vascular endothelial cells through blood flow and the ACh mediatesto vasodilation by forming and releasinge of NO from these cells [78], and the inhibitory effect of 50 Hz MFs on acetylcholinesterase (ACh lytic enzyme) has been reported in the magnetic intensity of 0.74 mT or more [39]. Furthermore, the modulatory effects of 50 Hz MFs on NO signaling pathways have been shown in the magnetic intensity of 1 mT (rms) in vitro [5, 40, 41]. For instance, $B_{rms}$ 1 mT 50 Hz MF for 3 h exposure significantly increased expression levels of induced NOS (iNOS) and endothelial NOS (eNOS) [5]. These 50 Hz MF-induced NOS expression increased in parallel with increased NOS activity [5]. Moreover, a recent clinical study revealed that a 40 Hz MF at 7 mT for 15 min/day significantly increased 3-nitrotyrosine and nitrate/nitrite levels in post-stroke patients, and improved functional and mental status [42]. Concerning the physiological mechanisms of NO, it has been reported that NO formed via eNOS and neuronal NOS (nNOS) causes vasodilatation, hypotension and blood flow increase [79]. Therefore, in this context, 50 Hz MF-induced NO activation/production could help promote blood circulation.

Considering future perspectives to understand the physiological mechanisms of 50 Hz MFs for hemodynamic responses, the induced EFs in three different regional exposures should be numerically calculated and analyzed in a more realistic model using the modified FEM method in more detail for precise microdosimetry purposes. Thus, computer simulations could open a new avenue for electromagnetically optimizing control of blood circulation, in particular, in ischemic regions. Moreover, objective assessment should be conducted for symptomatic patients with muscle fatigue and pain, in more severe cases, patients with ischemic disorders such as atherosclerosis and diabetes, to clarify the effects and mechanisms of 50 Hz MFs on ischemic regions. In addition, by exposure to 50 Hz MFs, there is a possibility that some specific biomarkers, such as melatonin, cortisol, serotonin and dopamine, could be affected [43, 44] and/or pain perception pathways via opioid receptors could be inhibited [45]. Further

studies should be needed to investigate the MF-based therapeutic applications and elucidate the underlying mechanisms of MF effects on pain relief and recovery of muscle fatigue.

### Limitations and future study needed

There are several limitations in this study. First, a primary limitation of this study relates to the lack of in vivo real-time measurement of NO production in the blood circulation under MF exposure. In addition to NO production [6, 40–42], the involvement of $Ca^{2+}$ [9–11, 13, 46–48] and ROS [14, 20, 49] in responses to 50 Hz MFs in vitro, which may also affect the FMD, has not been clarified yet in vivo. Moreover, direct real-time measurement of the amounts of the metabolic waste products and endogenous pain producing substances under MF exposure, will make the mechanism of MF effects more noticeable.

Another limitation of the present study relates to small sample size for a pilot study. To detect a statistical difference in the mean change from the baseline between real MF and sham control exposures, we re-estimated the magnitude of the MF effects on blood flow velocity for 15-min exposure in Fig 3 (also refer to Fig 2 and Table 2), by setting power = 0.80 using the above-mentioned EZR software package. Our obtained results of mean (SD) values of blood flow velocity differences between both conditions for 15-min exposure are 22.2% (9.4%), 12.7% (6.0%), and 11.6% (8.5%) in Fig 3A–3C, respectively. In these cases, the re-estimated sample size required to determine significant MF effects on blood flow velocity is at least 9 subjects. For the primary outcome measure, the re-estimated minimum sample size of 10 subjects would be able to detect significant differences between both conditions. In the case of FMD of 16 subjects in protocol B, mean (SD) values of FMD differences between both conditions were 1.59% (1.45%) for 30-min exposure in Fig 6. Here, the estimated minimum sample size is calculated to be 14 subjects per condition by setting power = 0.80 using the EZR software package. Therefore, we estimated that the total sample size of 16 subjects would be large enough to detect significant MF effects on FMD. We cannot rule out the possibility that differences between the MF and sham exposures may have become more significant if more subjects were studied.

Furthermore, the subjects are limited to normal healthy young men with normal vascular endothelial function. We examined men to avoid the cyclic effects of sex hormones on vascular endothelial function in women. Recruiting a large number of subjects would not have critically changed our findings. Thus, significant MF effects were found for a small number of healthy young men in this study, but further studies need to confirm whether MFs are effective in a larger number of older men and women.

### Conclusions

The ulnar artery blood flow velocity significantly increased corresponding to the $B_{max}$ values of MFs. These MF effects seem to be field strength-dependent and may be region-specific from the results of three different regional exposures to MFs. The induced EFs were highest underneath the coils, with minor diffusion to other regions. Furthermore, after muscle loading exercise, MF exposure recovered hemoglobin oxygenation index values faster and higher than sham exposure from the loading condition. The MF exposure significantly increased parasympathetic activity. The MF exposure significantly increased FMD values and the MF-induced increase of FMD values is directly related to the enhancement of NO production. These findings suggest that a 50 Hz MF can activate the parasympathetic nerve and induce hemodynamic responses via cholinergic mediators together with NO-mediated vasodilation. We believe that physiological monitoring results and computer simulations of the induced EFs provide new insight into how hemodynamic parameters can be changed in response to a 50 Hz MF at $B_{max}$

180 mT. The therapeutic implications of these findings require further preclinical and clinical investigations.

## Supporting information

**S1 File. Trial protocol and informed consent form.**
(DOC)

**S2 File. Trial protocol and informed consent form in Japanese.**
(DOC)

**S3 File. CONSORT checklist.**
(DOC)

**S4 File. Additional methods, results and figures.**
(DOCX)

**S5 File. Data set.**
(XLSX)

## Acknowledgments

We thank Ms. Megumi Hanada (UNEX Co., Ltd.) for her technical support and assistance with the FMD measurement.

## Author Contributions

**Conceptualization:** Hideyuki Okano.

**Data curation:** Hideyuki Okano.

**Formal analysis:** Hideyuki Okano, Akikatsu Fujimura, Tsukasa Kondo, Ilkka Laakso.

**Funding acquisition:** Keiichi Watanuki.

**Investigation:** Hideyuki Okano, Akikatsu Fujimura, Tsukasa Kondo, Ilkka Laakso.

**Methodology:** Hideyuki Okano, Akikatsu Fujimura, Tsukasa Kondo, Ilkka Laakso.

**Project administration:** Hideyuki Okano.

**Resources:** Hiromi Ishiwatari, Keiichi Watanuki.

**Software:** Ilkka Laakso.

**Supervision:** Hideyuki Okano.

**Validation:** Hideyuki Okano.

**Visualization:** Hideyuki Okano.

**Writing – original draft:** Hideyuki Okano, Ilkka Laakso.

**Writing – review & editing:** Hideyuki Okano, Ilkka Laakso.

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
