## [Decision Letter · Decision Letter 0]

26 Apr 2021

PONE-D-21-04188

A 50 Hz magnetic field affects hemodynamics, ECG and vascular endothelial function in healthy adults: A pilot randomized controlled trial

PLOS ONE

Dear Dr. OKANO,

Thank you for submitting your manuscript to PLOS ONE. After careful consideration, we feel that it has merit but does not fully meet PLOS ONE’s publication criteria as it currently stands. Therefore, we invite you to submit a revised version of the manuscript that addresses the points raised during the review process.

We look forward to receiving your revised manuscript.

Kind regards,

Yoshihiro Fukumoto

Academic Editor

PLOS ONE

Journal Requirements:

2. Thank you for submitting your clinical trial to PLOS ONE and for providing the name of the registry and the registration number. The information in the registry entry suggests that your trial was registered after patient recruitment began. PLOS ONE strongly encourages authors to register all trials before recruiting the first participant in a study.

1) your reasons for your delay in registering this study (after enrolment of participants started);

2) confirmation that all related trials are registered by stating: “The authors confirm that all ongoing and related trials for this drug/intervention are registered”.

3. We note that you attach your application form for Human Research Ethics Committee approval as your "Protocol" instead of the original written comprehensive protocol that would have been submitted along with the application for ethics review.  The protocol should be the complete and detailed plan for the conduct and analysis of the trial. Please send this in the original language. If this is in a language other than English, please also provide a translation. Please detail any deviations from this study protocol in the Methods section of your manuscript. Your study protocol will be made available to the editors and reviewers, and will be published as supporting information with your manuscript if accepted for publication. (If you do not agree to this, we will not be able to publish your manuscript). If you have formally published a study protocol for your trial in a journal then you should cite this in your manuscript, but you still need to send us the original document.

This study represents independent research part funded by the Advanced Institute of Innovative Technology, Saitama University. The funders had no role in study design, data collection and analysis, decision to publish, or preparation of the manuscript. There was no additional external funding received for this study.

5. Please include captions for ALL your Supporting Information files at the end of your manuscript, and update any in-text citations to match accordingly. Please see our Supporting Information guidelines for more information: http://journals.plos.org/plosone/s/supporting-information.

Reviewers' comments:

Reviewer's Responses to Questions

**Comments to the Author**

1. Is the manuscript technically sound, and do the data support the conclusions?

Reviewer #1: Partly

2. Has the statistical analysis been performed appropriately and rigorously? 

Reviewer #1: Yes

3. Have the authors made all data underlying the findings in their manuscript fully available?

Reviewer #1: No

4. Is the manuscript presented in an intelligible fashion and written in standard English?

Reviewer #1: Yes

5. Review Comments to the Author

Reviewer #1: PONE-D-21-04188: statistical review

SUMMARY. This study is a cross-over trial of the effects a magnetic field on hemodynamics, electrocardiogram, and vascular endothelial function in healthy young men. The core statistical analysis correctly relies on repeated-measures ANOVA. I however list below three specific issues that should be addressed.

SPECIFIC ISSUES

1. The Shapiro-Wilk test has been used for testing outcomes normality. The outputs of the tests are however not displayed. Please add these outputs as supplementary material: given the small sample size, checks of normality are crucial!

2. All the results are displayed in the form of figures, which is nice. However, the authors should also provide traditional tables with effects estimates, p-values and variance components. These tables would clarify the details of the statistical analysis and facilitate the replicability of the results.

2. Although the authors declare that the data are available without restrictions, the raw data are not attached to this submission. The data should be included as supplementary material or made available in a public repository.

6. PLOS authors have the option to publish the peer review history of their article (what does this mean?). If published, this will include your full peer review and any attached files.

Reviewer #1: No

---

## [Author Response · Author response to Decision Letter 0]

8 Jun 2021

Reply to Editor:

1. PLOS ONE’s style requirements.

We checked the PLOS ONE’s style requirements. 

2. Registration.

1) your reasons for your delay in registering this study (after enrolment of participants started).

Please see the methods:

The late registration was due to an error of omission. Such late registration does not affect study results and participants. We confirmed that all ongoing and related trials for this intervention are registered. We hereby state that all future trials will be registered prospectively.

2) confirmation that all related trials are registered by stating: “The authors confirm that all ongoing and related trials for this drug/intervention are registered”.

We confirmed that all trials for this intervention are registered. 

3. Protocol. 

The protocol should be the complete and detailed plan for the conduct and analysis of the trial. 

We addressed “Experimental protocol” attached “S1 file” in supporting information.

Please send this in the original language.

We addressed “Experimental protocol in Japanese” attached “S2 file” in supporting information.

4 Funding Statement.

We declared the statement “There was no additional external funding received for this study.” in our updated Funding Statement.

5. Please include captions for ALL your Supporting Information files at the end of your manuscript, and update any in-text citations to match accordingly.

We attached “S1-S5 files” as shown in supporting information at the end of our manuscript and updated the citations:

Supporting Information

S1 File. Trial protocol and informed consent form. 

(DOCX)

S2 File. Trial protocol and informed consent form in Japanese. 

(DOCX)

S3 File. CONSORT checklist.

(DOCX)

S4 File. Additional methods, results and figures.

(DOCX)

S5 File. Data set.

(XLSX)

Reply to Reviewer: 

1. Analysis of the Shapiro-Wilk test.

We showed all of the outputs of the Shapiro-Wilk tests as “The Shapiro-Wilk test results for normality assumptions” in “Table A in S4 File”:

Supporting Information

S4 File. Additional methods, results and figures.

(DOCX)

Table A. The Shapiro-Wilk test results for normality assumptions.

1. Subject baseline characteristics in the protocol A and B.

2. Comparison of change rate (%) of ulnar arterial blood flow velocity (BFV), blood pressure (BP), heart rate (HR), hemoglobin oxygenation index (HOI), RRI, HF, LF/HF and FMD between 50 Hz MF and sham exposures.

2. Presentation of the tables indicating p-values and variance components. 

We showed all of the tables with p-values in the results:

Table 3. Comparison of change rate (%) of ulnar arterial blood flow velocity between 50 Hz MF and sham exposures. 

Table 4. Comparison of blood pressure (BP) and heart rate (HR) between 50 Hz MF and sham exposures in the forearm.

Table 5. Comparison of change rate (%) of hemoglobin oxygenation index between 50 Hz MF and sham exposures in the forearm. 

Table 6. Comparison of change rate (%) of ECG parameters between 50 Hz MF and sham exposures in the neck.

Table 7. Comparison of FMD (%) between 50 Hz MF and sham exposures in the upper arm. 

3. Presentation of the raw data.

We attached all of the raw data as shown in “S5 File”:

Supporting Information

S5 File. Data set.

(XLSX)

---

## [Decision Letter · Decision Letter 1]

13 Jul 2021

A 50 Hz magnetic field affects hemodynamics, ECG and vascular endothelial function in healthy adults: A pilot randomized controlled trial

PONE-D-21-04188R1

Dear Dr. OKANO,

We’re pleased to inform you that your manuscript has been judged scientifically suitable for publication and will be formally accepted for publication once it meets all outstanding technical requirements.

Kind regards,

Yoshihiro Fukumoto

Academic Editor

PLOS ONE

Additional Editor Comments (optional):

Reviewers' comments:

Reviewer's Responses to Questions

**Comments to the Author**

1. If the authors have adequately addressed your comments raised in a previous round of review and you feel that this manuscript is now acceptable for publication, you may indicate that here to bypass the “Comments to the Author” section, enter your conflict of interest statement in the “Confidential to Editor” section, and submit your "Accept" recommendation.

Reviewer #1: All comments have been addressed

2. Is the manuscript technically sound, and do the data support the conclusions?

Reviewer #1: (No Response)

3. Has the statistical analysis been performed appropriately and rigorously? 

Reviewer #1: (No Response)

4. Have the authors made all data underlying the findings in their manuscript fully available?

Reviewer #1: (No Response)

5. Is the manuscript presented in an intelligible fashion and written in standard English?

Reviewer #1: (No Response)

6. Review Comments to the Author

Reviewer #1: (No Response)

7. PLOS authors have the option to publish the peer review history of their article (what does this mean?). If published, this will include your full peer review and any attached files.

Reviewer #1: No

---

## [Editor Report · Acceptance letter]

19 Jul 2021

PONE-D-21-04188R1 

A 50 Hz magnetic field affects hemodynamics, ECG and vascular endothelial function in healthy adults: A pilot randomized controlled trial 

Dear Dr. Okano:

I'm pleased to inform you that your manuscript has been deemed suitable for publication in PLOS ONE. Congratulations! Your manuscript is now with our production department. 

Kind regards, 

on behalf of

Dr. Yoshihiro Fukumoto 

Academic Editor

PLOS ONE